# BNIP3L/NIX-mediated mitophagy protects against glucocorticoid-induced synapse defects

Gee Euhn Choi[1], Hyun Jik Lee[2,3], Chang Woo Chae[1], Ji Hyeon Cho[1], Young Hyun Jung[1], Jun Sung Kim [1], Seo Yihl Kim[1], Jae Ryong Lim[1] & Ho Jae Han [1✉]

Stress-induced glucocorticoids disturb mitochondrial bioenergetics and dynamics; however, instead of being removed via mitophagy, the damaged mitochondria accumulate. Therefore, we investigate the role of glucocorticoids in mitophagy inhibition and subsequent synaptic defects in hippocampal neurons, SH-SY5Y cells, and ICR mice. First, we observe that glucocorticoids decrease both synaptic density and vesicle recycling due to suppressed mitophagy. Screening data reveal that glucocorticoids downregulate BNIP3-like (BNIP3L)/NIX, resulting in the reduced mitochondrial respiration function and synaptic density. Notably, we find that glucocorticoids direct the glucocorticoid receptor to bind directly to the PGC1α promoter, downregulating its expression and nuclear translocation. PGC1α downregulation selectively decreases NIX-dependent mitophagy. Consistent with these results, NIX enhancer pre-treatment of a corticosterone-exposed mouse elevates mitophagy and synaptic density in hippocampus, improving the outcome of a spatial memory task. In conclusion, glucocorticoids inhibit mitophagy via downregulating NIX and that NIX activation represents a potential target for restoring synapse function.

[1] Department of Veterinary Physiology, College of Veterinary Medicine, Research Institute for Veterinary Science, and BK21 Four Future Veterinary Medicine Leading Education & Research Center, Seoul National University, Seoul 08826, South Korea. [2] Laboratory of Veterinary Physiology, College of Veterinary Medicine, Chungbuk National University, Cheongju, Chungbuk 28644, Korea. [3] Institute for Stem Cell & Regenerative Medicine (ISCRM), Chungbuk National University, Cheongju, Chungbuk 28644, South Korea. ✉email: hjhan@snu.ac.kr

Stress-induced increases in levels of glucocorticoids, a major etiology of neurodegenerative diseases such as Alzheimer's disease (AD)[1], also trigger mitochondrial damage such as arrested motility, fission via Drp1 phosphorylation, impaired $Ca^{2+}$ buffering, and excessive mitochondrial reactive oxygen species (mtROS) production[2]. In normal neurons, damaged mitochondria undergo asymmetrical fission and are targeted to the autophagosome, followed by return to the soma for degradation in lysosomes or removal at distal axons[3,4]. This process called mitophagy maintains an appropriate number of healthy organelles in the mitochondrial pool particularly at synapses where most ATP is required. However, the detachment of dysfunctional mitochondria from the axonal microtubules and their accumulation around the nucleus is observed in neurons under high levels of glucocorticoids[5]. Furthermore, epinephrine, another stress-induced factor, reportedly inhibits autophagy, which can contribute to mitochondrial dysfunction and neurodegenerative diseases such as AD[6,7]. Failure to eliminate damaged mitochondria in stress-exposed neurons induces abnormality in synaptic homeostasis and ultimately promotes neurodegeneration. Therefore, it can be assumed that appropriate activation of mitophagy machinery is hampered by glucocorticoids; however, the detailed mechanism remains unclear.

Upon mitochondrial membrane depolarization, the PTEN-induced kinase 1 (PINK1)-parkin pathway is activated to ameliorate several neurodegenerative diseases[8]. However, many studies oppose this paradigm because the physiological level of parkin is insufficient to induce mitophagy[9–11]. Specifically, mitophagy mediated by the PINK1-parkin pathway is largely unnecessary for regulating basal mitophagy, which refers to continuous mitochondrial housekeeping under normal state or chronic disease conditions in neural tissues[12]. Rather, the PINK1-parkin pathway compensates for acute, chemical insult-mediated mitochondrial dysfunction[12]. However, PINK1-independent, receptor-mediated mitophagy is strongly associated with basal mitophagy, which accounts for the largest number of mitophagy processes and is more important in energy-demanding tissues including brain[13]. In fact, patients with AD or cerebral ischemia exhibit suppression of NIX or FUN14 domain containing 1-mediated mitophagy with no changes in PINK1-parkin pathway[14,15]. This indicates that receptor-mediated mitophagy operates high rate under chronic stress situations[9,11]. Because stress-mediated neurodegeneration is chronic and potentially affects the level of basal mitophagy, we surmise that high concentrations of glucocorticoids induced by moderate to severe stress impair basal mitophagy; restoring this function may recover synaptic damage and neuronal cell viability.

In the present study, mouse primary hippocampal neurons and human neuroblastoma SH-SY5Y cells, common in vitro neurodegenerative models, were used to investigate the detailed mechanism of mitophagy dysfunction by chronic exposure to high levels of corticosterone and cortisol. In addition, we used ICR mice to stress-induced levels of corticosterone to assess how it can affect mitophagy and subsequent synapse/spatial memory impairment. Overall, the study addresses the effect of stress-induced levels of glucocorticoids on basal mitophagy machinery and ensuing neurodegeneration including synapse or cognitive impairments using both in vitro and in vivo models.

## Results
**Glucocorticoids induce synaptic dysfunction resulting from defective mitophagy processes.** Hippocampus, which regulates memory and mood, is the brain region most responsive to glucocorticoids. Therefore, we performed hippocampal neuron culture throughout this study. Corticosterone and cortisol are the major glucocorticoids released under stress conditions in rodents and humans, respectively. The physiological plasma concentrations of glucocorticoids is relatively low, roughly between 7 and 35 ng/ml. Stress-induced glucocorticoid levels are ~70 ng/ml or higher[16]. Low glucocorticoid levels usually activate mineralocorticoid receptor (MR)-mediated responses, whereas high glucocorticoid levels predominantly induce glucocorticoid receptor (GR)-dependent signaling[17]. Glucocorticoids display both circadian and ultradian rhythms, with oscillation peaks around Zeitgeber time (ZT) 13. These rhythms maintain the transcriptional activity of glucocorticoids to sustain normal levels under neuronal homeostasis, primarily mediated by the MR even at ZT 13[18]. However, stress-induced increase in glucocorticoids activates the stress response, not blocked by MR antagonists. Corticosterone levels in mice under stress increase up to fivefold compared with control mice. These mice showed corticosterone concentrations of 300–500 ng/ml after restraint stress, approximately equivalent to 1 μM of corticosterone[19]. Thus, 1 μM corticosterone and cortisol are widely used for inducing a stress response in vitro[20]. Furthermore, 100 nM corticosterone and cortisol, ~35 ng/ml, is considered equivalent to low, physiological levels. To analyze the dose-dependent effect of corticosterone and cortisol on mitophagy inhibition, we performed the western blot to detect the levels of the mitochondrial marker TOMM20. We confirmed that 1 μM corticosterone and cortisol inhibited mitophagy in hippocampal neurons and SH-SY5Y cells, respectively (Supplementary Fig. 1a, b). To assess synaptic density, Pearson's correlation coefficient between synapse markers (synaptophysin and PSD95) are analyzed in that synaptophysin is a presynaptic vesicle protein while PSD95 is a postsynaptic scaffold protein. We also observed the dose-dependent effect of corticosterone on synaptic density. Only 1 μM corticosterone was sufficient to reduce synaptic density (Supplementary Fig. 1c). Therefore, we treated neuronal cells with 1 μM corticosterone and cortisol for >24 h in all subsequent in vitro experiments, which mimics chronic elevation of glucocorticoids to evaluate the effect of stress-induced glucocorticoid levels on mitophagy.

In hippocampal neurons, most mitochondria are normally localized in distal dendrites or axonal processes[21]. Thus, we evaluated the mitochondrial distribution in distal neurites via staining MAP2 for dendrites, Tau for axons, and mitotracker red (MTR) for mitochondria. The length of distal neurites is commonly set to ~100 μm[22]. We determined the distribution of mitochondria in distal dendrites and distal axons by measuring colocalization between MAP2 and MTR and colocalization between Tau and MTR, respectively. Corticosterone redistributed mitochondria away from dendrites and axons, but it had no effect on Tau expression or distribution (Fig. 1a). Damaged mitochondria are typically transported to the soma in a retrograde fashion to fuse with lysosomes and degradation. Repaired or newly generated healthy mitochondria are then transported to distal neurons to maintain the synapse homeostasis[23]. Failure to clear the damaged mitochondria in soma triggers loss of distal mitochondria and perinuclear clumping of abnormal mitochondria, followed by shortening of dendritic lengths[24]. Therefore, we immunostained the hippocampal neurons with antibodies recognizing the mitochondria marker TOMM20, the synaptic marker synaptophysin, and dyed nuclear DNA (nDNA) with the stain DAPI. We measured the intensity of TOMM20 immunofluorescence from the nucleus up to a distance of 20 μm (perinuclear region). And we evaluated the Pearson's correlation coefficient between synaptophysin and TOMM20 to observe mitochondrial localization at synapse. Corticosterone evoked accumulation of mitochondria around the nucleus, rather than the synaptic region (Fig. 1b). We also set the perinuclear region and distal region ~20 μm from nucleus and cell extremities,

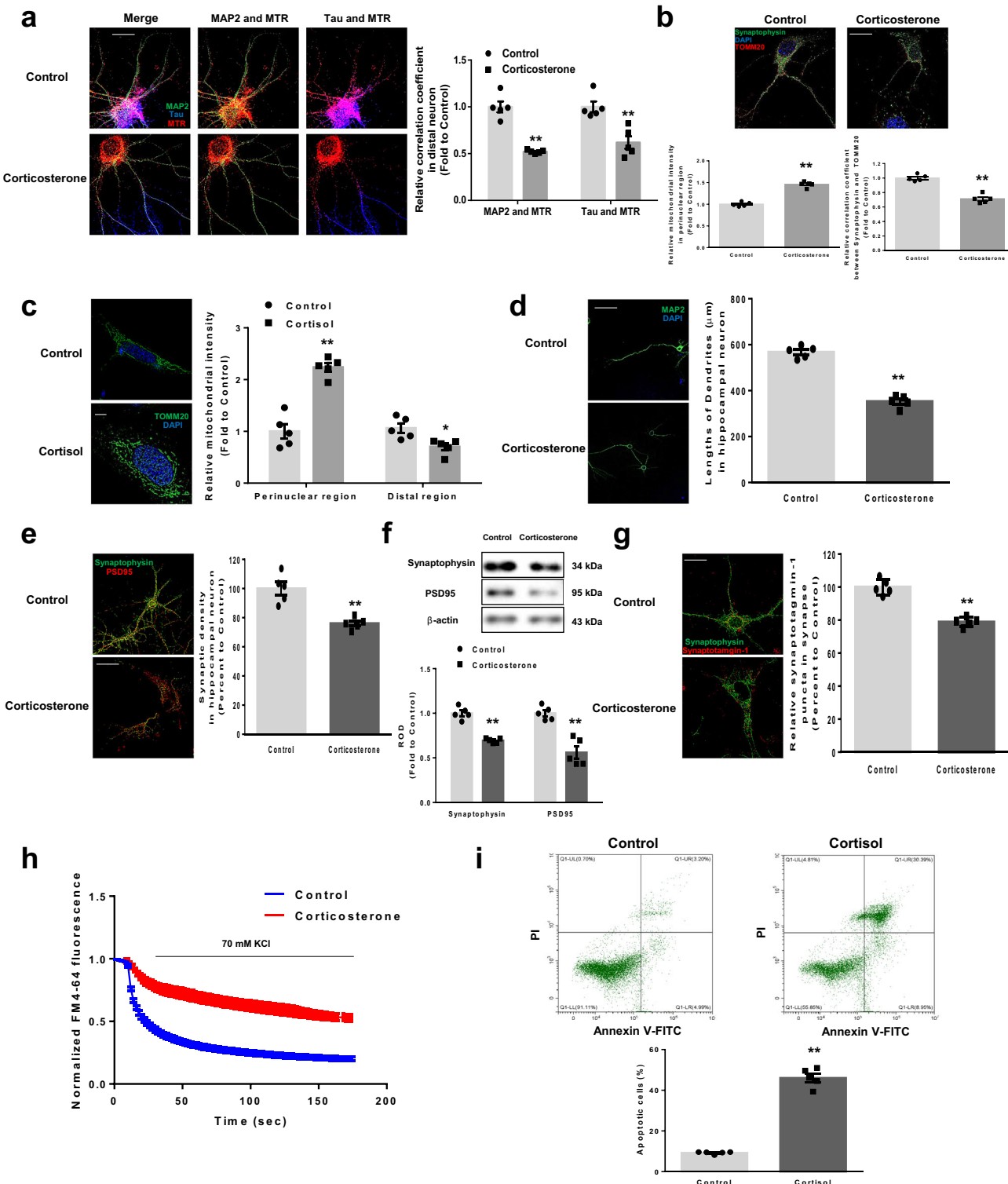

respectively, in SH-SY5Y cells. Then we evaluated the mitochondrial intensity in both areas. Similarly, mitochondria accumulated around the nucleus rather than extremities by cortisol (Fig. 1c). Mitochondrial dysfunction and mislocalization eventually lead to synaptic abnormalities owing to deficiencies in ATP production and insufficient $Ca^{2+}$ buffering[25]. We found that corticosterone shortened dendritic lengths (Fig. 1d), decreased synaptic density (Fig. 1e), and reduce levels of synapse markers (Fig. 1f). Owing to decreased synaptic function, hippocampal neurons under corticosterone displayed reduced synaptic vesicle recycling, as examined by a synaptotagmin uptake assay (Fig. 1g). In addition, corticosterone repressed the rate of FM4-64 dye release, representing that recycling synaptic vesicle pool size and subsequent synaptic currents were decreased (Fig. 1h). SH-SY5Y cells also underwent significant apoptosis under cortisol (Fig. 1i).

Damaged mitochondria normally undergo mitophagy and disappear around nucleus while healthy mitochondria are transported toward the neuronal terminus. Thus, we examined the effect of corticosterone and cortisol on mitophagy in both

**Fig. 1 Glucocorticoids induce mitochondrial mislocalization, synaptic dysfunction, and cell death. a–c** Hippocampal neurons and SH-SY5Y cells were treated with corticosterone and cortisol for 24 h, respectively. Hippocampal neurons were immunostained with MAP2 (green), Tau (blue), and Mitotracker red (MTR, red). Relative ratio of Pearson's correlation coefficient was quantified in distal neurites (100 μm). Scale bars, 20 μm (magnification, ×1000). n = 5. **b** Hippocampal neurons were immunostained with synaptophysin (green), DAPI (blue), and TOMM20 (red). Mitochondrial intensity around perinuclear regions and Pearson's correlation coefficient between synaptophysin and TOMM20 were quantified. Scale bars, 100 μm (magnification, ×200). n = 5. **c** SH-SY5Y cells were immunostained with TOMM20 (green) and DAPI (blue). Mitochondrial intensity was quantified around nucleus and extremities. Scale bars, 20 μm (magnification, ×1000). n = 5. **d–h** Hippocampal neurons were treated with corticosterone for 48 h. **d** Hippocampal neurons were immunostained with MAP2 (green) and DAPI (blue) for measuring lengths of dendrites. Scale bars, 100 μm (magnification, ×200). n = 5. **e** Hippocampal neurons were immunostained with synaptophysin (green) and PSD95 (red). Pearson's correlation coefficient was quantified for detecting synaptic density. Scale bars, 100 μm (magnification, ×200). n = 5. **f** Western blot was performed. n = 5. **g** After stimulation with depolarization buffer, cells were incubated with synaptotagmin-1 antibody and then fixed. Synaptophysin (green) was used to visualize synapse and synaptotagmin-1 (red) was used for measuring uptake of synaptic vesicles. Scale bars, 100 μm (magnification, ×200). n = 5. **h** Conditioned hippocampal neurons were stained with FM4-64 dye and stimulated with high K⁺ buffer for destaining. Time-lapse imaging was done over 180 sec at 1 sec intervals. n = 5. **i** SH-SY5Y cells were treated with cortisol for 72 h. The percentages of apoptotic cells (Annexin V positive cells) were analyzed by Annexin V/PI analysis, measured by flowcytometer. n = 5. All blot and immunofluorescence images are representative. n = 5 from independent experiments with two technical replicates each. Quantitative data are presented as a mean ± S.E.M. The representative images were acquired by SRRF imaging system. Two-sided unpaired student's $t$ test was conducted. *, ** indicates $p < 0.05$, $p < 0.01$ versus control, respectively. Data are provided as a Source data file.

cells. Mitophagy reporter mitochondria-targeted Keima (mt-Keima) is a good indicator to evaluate mitophagic flux. Mt-Keima is a coral-derived acid-stable protein, which changes color from green to red when mitophagosomes are under acidic conditions in lysosomes[26]. Cells were treated with carbonyl cyanide 3-chlorophenylhydrazone (CCCP) and antimycin A1, which pharmacologically induce mitophagy by disrupting mitochondrial membrane potential and inhibiting complex III of mitochondrial electron transport chain, respectively, as positive controls for mitophagy induction. Corticosterone and cortisol reduced the ratio of red signal to the sum of red and green mt-Keima fluorescence in both cell types, suggesting they significantly inhibited mitophagy (Fig. 2a, b). Following corticosterone and cortisol treatment, both cell types also displayed reduced colocalization between LC3 and mitochondria, as determined using AutophagSense (representing LC3, which bridges mitophagosome components with dysfunctional mitochondria) and DsRed2-mito plasmid transfection (Fig. 2c, d). We measured the amount of mitochondrial DNA (mtDNA) impairment, occurring under mitophagy inhibition, using a terminal deoxynucleotidyl transferase dUTP nick end labeling (TUNEL) assay co-stained with MTR. Mitochondrial clearance was reduced under corticosterone and cortisol treatment (Fig. 2e, f). Using a mtDNA damage assay[27], corticosterone, and cortisol were found to negatively affect mtDNA clearance (Fig. 2g, h). However, mtDNA levels may also increase as a result of increased mitochondrial biogenesis. Mitochondrial biogenesis is the cellular process by which new mitochondria are synthesized from the growth and division of pre-existing mitochondria in adapting to the energy requirements. When mitochondria become aged or defective, organelle clearance occurs primarily through mitophagy, typically followed by biogenesis under homeostatic conditions[28]. We measured whether cortisol affected mitochondrial biogenesis using MitoBiogenesis™ In-Cell ELISA kit. Two proteins were assessed: subunit I of Complex IV (COXI), a mtDNA-encoded protein, and the 70 kDa subunit of Complex II (SDH-A), a nuclear-encoded protein. Mitochondrial biogenesis and content are quantified as a measure of protein synthesis of COXI relate to SDH-A. Cortisol decreased mitochondrial biogenesis, suggesting that the observed increase in mtDNA did not occur as a result of an increase in mitochondrial biogenesis (Supplementary Fig. 2a). From this observation, we conclude that it is suitable to measure mtDNA levels for detecting mitochondrial contents, which was mainly increased by failure to eliminate dysfunctional mitochondria in our experimental conditions. We also analyzed mitophagic flux in hippocampal neurons using bafilomycin A1, which blocks the fusion between mitophagosomes and lysosomes, resulting in accumulation of mitophagosomes[29]. Corticosterone increased mitochondrial contents. And bafilomycin A1 alone increased TOMM20 levels and the LC3II/LC3I ratio, but when co-treated with corticosterone, did not significantly alter the LC3II/LC3I ratio (Fig. 2i). These data indicated that mitophagosome formation is inhibited by corticosterone. Mitophagosomes can be degraded by transcellular mitophagy rather than retrograde to the lysosomes in neuronal axons[21]. To determine whether corticosterone inhibited mitophagy via the suppression of transcellular mitophagy, we examined the levels of γ-synuclein, a byproduct of astrocyte phagocytosis of axonal elements, in hippocampal neurons[30]. No significant changes in γ-synuclein levels were detected (Supplementary Fig. 2b). We surmised that despite research that suggests that local mitophagy was observed in distal axons or dendrites, mitophagic events primarily occur in the soma with limited events in either dendrites or axons[21,31]. Collectively, these findings indicated that corticosterone and cortisol have a significant inhibitory effect on mitophagy, resulting in impaired mitochondrial distribution and synaptic homeostasis.

**NIX reduction by glucocorticoids independent from the PINK1-parkin pathway is a key pathogenic factor for mitochondrial and synaptic dysfunction.** To investigate which factor among the mitophagy regulators is decreased upon cortisol treatment, we checked mRNA expressions. As shown, the expression of *NIX* mRNA was decreased, whereas that of *BNIP3* and *PINK1* remained unchanged (Fig. 3a). Consistently, we found that corticosterone and cortisol selectively downregulated NIX in both hippocampal neurons and SH-SY5Y cells, respectively (Fig. 3b, c). As a result, the existence of NIX in mitochondria was decreased upon corticosterone and cortisol treatment, as revealed by immunostaining (Fig. 3d, e) and mitochondrial fraction (Supplementary Fig. 3a). Several studies demonstrated that NIX function in certain tissues is dependent on parkin-mediated signaling pathway[32]. We observed that cortisol decreased the binding between parkin and NIX, but *PARK2* knockdown had no effect on further NIX downregulation (Fig. 3f). Furthermore, we determined whether parkin upregulation would recover NIX expression. We transfected hippocampal neurons and SH-SY5Y cells with mt-Keima-Red-Parkin, which co-express mt-Keima and parkin, known to play a role in overexpression of parkin. Unlike the previous reports, we confirmed that NIX expression was solely decreased by corticosterone and cortisol whether or not parkin protein was abundant (Fig. 3g, h). We demonstrated

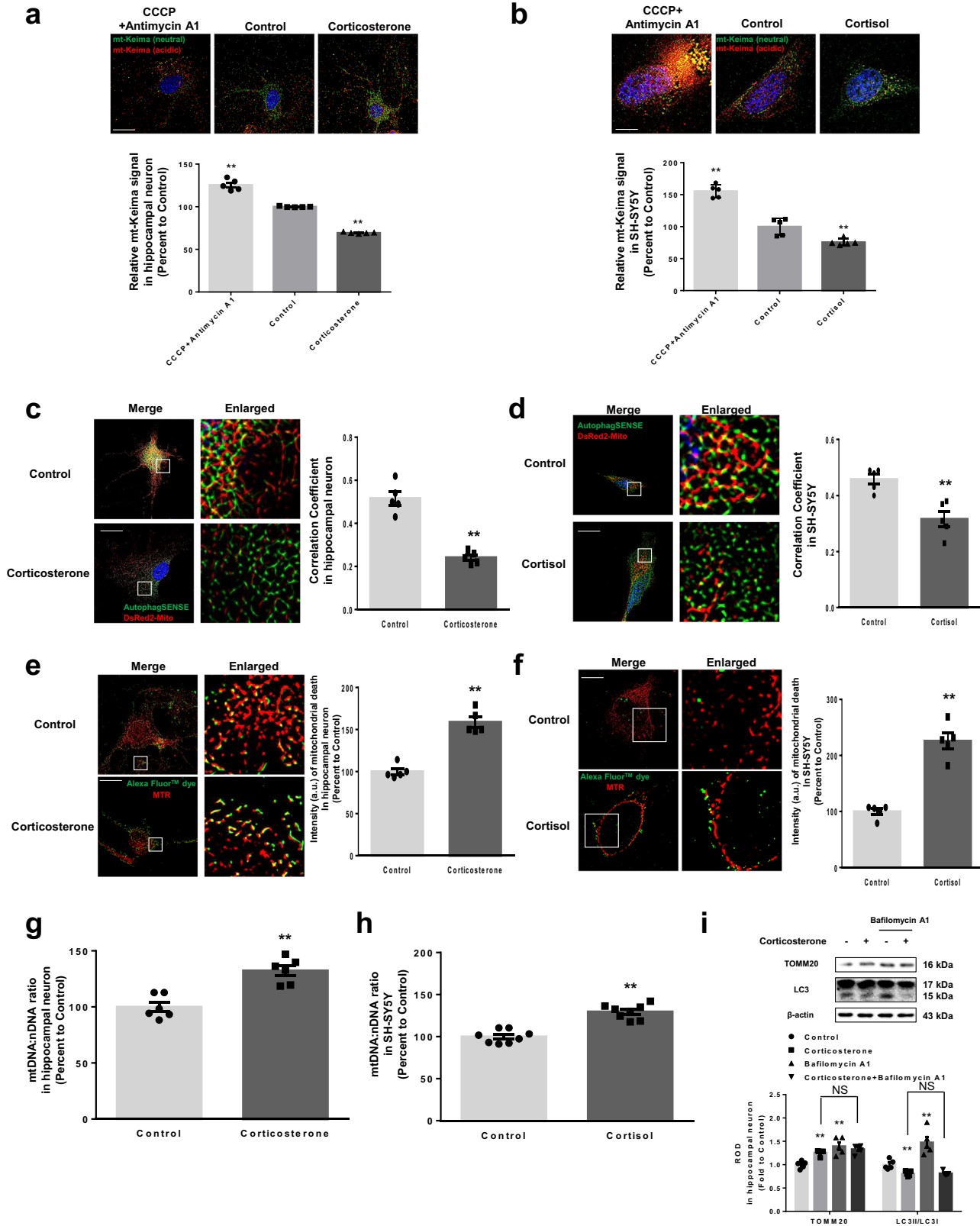

that the activation of PINK1-parkin pathway by antimycin A1 did not affect mitophagy levels under cortisol treatment but NIX enhancement by phorbol 12-myristate 13-acetate (PMA) recovered mitophagy levels (Supplementary Fig. 3b)[33]. These data suggested that corticosterone and cortisol strongly inhibit NIX-mediated mitophagy, not affecting PINK1-parkin pathway. We

also performed immunoprecipitation and immunocytochemistry to detect the ubiquitination of mitochondria, the primary downstream event of PINK1-parkin signaling. Ubiquitination of TOMM20 was not altered by cortisol treatment (Supplementary Fig. 3c). Furthermore, the interaction between ubiquitin and Mfn1, another PINK1-parkin substrate, was unchanged upon

**Fig. 2 Corticosterone and cortisol suppress mitophagy. a, b** Hippocampal neurons were transfected with mitophagy reporter mitochondria-targeted Keima (mt-Keima) at DIV 5 whereas SH-SY5Y cells were transfected with mt-Keima 48 h prior to treatment. Carbonyl cyanide 3-chlorophenylhydrazone (CCCP, 10 μM) and antimycin A1 (1 μM) were pretreated for 2 h. And corticosterone and cortisol were treated for 24 h in hippocampal neurons at DIV 14 and SH-SY5Y cells, respectively. Green and red fluorescence of mt-Keima were visualized. DAPI was used for nuclear counterstaining (blue). Ratio of red to sum of red and green was quantified. Scale bars, 20 μm (magnification, ×1000). $n = 5$. **c, d** Hippocampal neurons were transfected with Autophagsense (GFP) and DsRed2-Mito (RFP) at DIV 5. SH-SY5Y was transfected with two same vectors 48 h prior to treatment. Corticosterone and cortisol were treated for 24 h in hippocampal neurons at DIV 14 and SH-SY5Y cells, respectively. Scale bars, 20 μm (magnification, ×1000). $n = 5$. Pearson's correlation coefficient between GFP and RFP fluorescence was quantified. **e-i** Hippocampal neurons and SH-SY5Y cells were treated with corticosterone and cortisol for 24 h, respectively. **e, f** Both cell types were stained with MTR (red) for 30 min and then fixed with formaldehyde. TUNEL assay was subsequently done. DNA breaks were detected with FITC filter. FITC fluorescence in mitochondria was quantified as mitochondrial death. Scale bars, 20 μm (magnification, ×1000). $n = 5$. **g, h** DNA levels of mitochondrial DNA (mtDNA) and nDNA were analyzed by real-time PCR. *ACTB* was used as a loading control. $n = 5$. **i** Bafilomycin A1 (10 nM) was applied for 2 h prior to harvest. TOMM20 expression and LC3II/I ratio in hippocampal neuron were analyzed by western blot. Loading control is β-actin. $n = 5$. All blots and immunofluorescence images are representative. $n = 5$ from independent experiments with two technical replicates each. Quantitative data are presented as a mean ± S.E.M. The representative images were acquired by SRRF imaging system. Two-sided unpaired student's *t* test was conducted except Figs. 2a, b and 2i, data of which were analyzed by two-way ANOVA. [**] indicates $p < 0.01$ versus control. NS means not significant. Data are provided as a Source data file.

cortisol treatment (Supplementary Fig. 3d). To confirm that PMA had an effect on NIX expression, we used *BNIP3L* small interfering RNA (siRNA). Knockdown of *BNIP3L* abolished the enhancing effect of PMA on NIX expression and even decreased its expression compared to that of the control (Supplementary Fig. 3e). In reality, PMA recovered mitophagy levels in cells under corticosterone and cortisol treatment, analyzed by detecting mt-Keima signals and TOMM20 levels (Fig. 3i–l). In SH-SY5Y cells, NIX overexpression reversed the increased effect of TOMM20 by cortisol (Fig. 3m). Collectively, our results suggested that NIX downregulation by corticosterone and cortisol mainly inhibits mitophagy independently from the parkin-mediated pathway in neuronal cells.

Next, we investigated whether mitochondrial and synaptic dysfunction were recovered by NIX upregulation. We used mitoSOX Red, a cell permeant mitochondrial superoxide indicator, to measure mtROS levels. Increased mtROS were decreased by PMA and NIX overexpression in hippocampal neurons and SH-SY5Y cells, respectively (Fig. 4a, b). Tetramethylrhodamine ethyl ester (TMRE) intensity, a measure used to quantify mitochondrial membrane potential, was recovered by NIX upregulation (Fig. 4c, d). Excessive oxidative stress and decreased mitochondrial membrane potential ultimately leads to dysfunction of mitochondrial biogenesis. Cortisol decreased mitochondrial biogenesis represented by reduced COXI/SDH-A ratio, but NIX upregulation reversed this effect (Supplementary Fig. 4a, b). Also, oxygen consumption rate (OCR) data showed that corticosterone and cortisol decreased basal respiration, maximal respiration, ATP production, and proton leak. However, NIX upregulation ameliorated the results (Fig. 4e–h). Decreased mitochondrial function is a major early event of synaptic dysfunction and apoptosis. We determined that the reduction in synaptic markers and vesicle recycling were normalized to a control level following PMA pretreatment of hippocampal neurons (Fig. 4i–k). Furthermore, the decreased dendritic lengths in hippocampal neuron and apoptosis of SH-SY5Y cells were recovered by NIX upregulation (Supplementary Fig. 4c, d). Overall, the damaging effects of corticosterone and cortisol on mitochondrial function and synapse homeostasis were well restored with NIX upregulation.

**GR represses PGC1α-NIX axis leading to mitophagy impairment.** To uncover the molecular causes of NIX downregulation, we identified a contribution by GR, which is activated mainly by relatively high levels of glucocorticoids, to the inhibition of mitophagy. In both hippocampal and SH-SY5Y cells, the knockdown of *GR* recovered mitophagy levels according to the results from mt-Keima signals (Fig. 5a, b) and TOMM20 levels (Fig. 5c, d). In addition, we demonstrated that the knockdown of *GR* recovered NIX expression in both cell types under corticosterone and cortisol (Fig. 6a, b). To determine which signaling pathway is dominant in NIX expression, we investigated the genes associated with nuclear receptors and coregulators. PCR array data revealed that cortisol reduced expressions of genes including *HDAC1/2*, *NR2C2*, *NR3C1* (*GR*), *PPARGC1A*, *MED12/14/17*, and *NOTCH2*. Among these genes, both peroxisome proliferator-activated receptor gamma coactivator 1-alpha (PGC1α) and GR expression were significantly decreased (Fig. 6c). Several hours of glucocorticoid exposure is known to decrease GR expression for negative feedback so that *NR3C1* is reasonable to be reduced. Among the downregulated genes, we surmised that PGC1α is a strong candidate for NIX regulation because promoter of this gene has more GR-binding regions than other downregulated genes. There are several GR-binding motifs, known as glucocorticoid response element including TGTTCT sequence, in PGC1α promoter sites, −1000 bp upstream of the first codon of the gene (Fig. 6d). PGC1α is known to be a potential therapeutic target in many neurodegenerative diseases, owing to its activating role in mitophagy-related pathways[34]. Hence, it was likely that GR would repress PGC1α action, resulting in mitophagy inhibition. Chromatin immunoprecipitation (ChIP) assay results also supported the cortisol-induced binding of GR to the PGC1α promoter, indicating that ligand-bound GR could trans-repress PGC1α expression (Fig. 6e).

Next, we evaluated changes in the expression and localization of PGC1α. We found that knockdown of *GR* ameliorated PGC1α repression by corticosterone and cortisol (Fig. 7a, b). We also detected that nuclear translocation of PGC1α was significantly reduced by corticosterone and cortisol, which was increased by *GR* knockdown, as observed in immunostaining and subcellular fraction results (Fig. 7c–e). Consistent with these results, PGC1α overexpression eventually recovered NIX expression, suggesting that the GR-PGC1α axis is specific to NIX regulation (Fig. 7f). Finally, the recovery of NIX by PGC1α upregulation strongly rescued mitophagy inhibition by cortisol (Fig. 7g). Therefore, corticosterone and cortisol mitigated NIX-dependent mitophagy in both hippocampal and SH-SY5Y cells by inducing GR-dependent signaling, which trans-repressed PGC1α gene expression.

**NIX upregulation improved synapse and cognitive function in corticosterone-exposed mouse.** With the pathway having been established, we performed in vivo experiments to explore the neurodegeneration phenotypes such as behavior changes by high

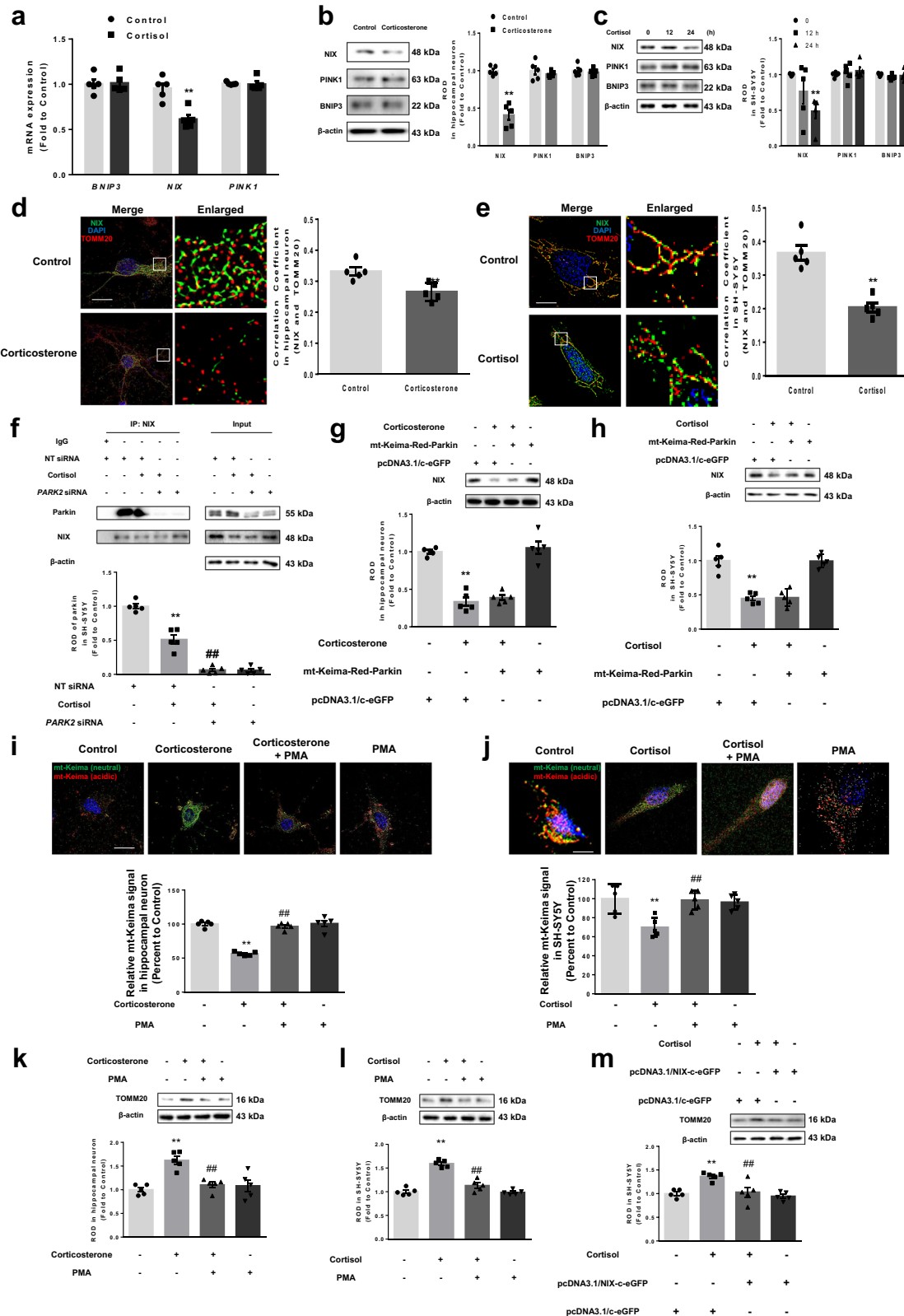

doses of corticosterone (10 mg/kg) following the drug regime (Supplementary Fig. 5a). To assess mitophagy using immunostaining, we stained the brain tissue with LAMP1 and TOMM20 antibody for determining the fusion of lysosome and mitochondria, which form mitolysosome. NIX upregulation by PMA increased colocalization between LAMP1 and TOMM20, meaning

that PMA pretreatment attenuated the suppressive effect of corticosterone on mitophagy (Fig. 8a). We then confirmed that NIX was selectively downregulated by corticosterone (Fig. 8b). Corticosterone also reduced synaptic density and marker levels in hippocampus, which were recovered by PMA pretreatment (Fig. 8c, d). Decreased synaptic plasticity and function in

**Fig. 3 Glucocorticoids suppress NIX expression independent from PINK1-parkin pathway. a** SH-SY5Y cells were incubated with cortisol for 12 h and mRNA expressions were analyzed by real-time PCR. $n = 5$. **b, c** Hippocampal neurons and SH-SY5Y cells were treated with corticosterone for 24 h and with cortisol for various time, respectively. Expression of NIX, PTEN-induced kinase 1 (PINK1), and BCL2 interacting protein 3 (BNIP3) were detected via western blot. $n = 5$. **d, e** Hippocampal neurons and SH-SY5Y cells were treated with corticosterone and cortisol, respectively, for 24 h. NIX (green), TOMM20 (red), and DAPI (nuclear counterstaining, blue) were visualized. Scale bars, 20 μm (magnification, ×1000). $n = 5$. **f** Nontargeting (NT) or *PARK2* siRNA was transfected to SH-SY5Y cells for 24 prior to cortisol for 24 h. NIX was co-immunoprecipitated with parkin. Parkin levels in immunoprecipitated samples were quantified. $n = 5$. **g, h** Both hippocampal neurons and SH-SY5Y cells were transfected with mitochondria-targeted Keima (mt-Keima)-Red-Parkin for 24 h prior to corticosterone and cortisol for 24 h, respectively. Western blot was performed. $n = 5$. **i–l** Both hippocampal neurons and SH-SY5Y cells were pretreated with phorbol 12-myristate 13-acetate (PMA, 10 nM) for 30 min before corticosterone and cortisol for 24 h, respectively. **i, j** Hippocampal neurons and SH-SY5Y cells were transfected with mt-Keima at DIV 5 and 48 h prior to treatment, respectively. DAPI was used for nuclear counterstaining (blue). Ratio of red to sum of red and green was quantified. Scale bars, 20 μm (magnification, ×1000). $n = 5$. **k, l** TOMM20 levels were detected by western blot. $n = 5$. **m** SH-SY5Y cells were transfected with pcDNA3.1/c-eGFP or pcDNA3.1/NIX-c-eGFP vector for 24 h prior to cortisol for 24 h. TOMM20 levels were detected by western blot. $n = 5$. All blots and immunofluorescence images are representative. $n = 5$ from independent experiments with two technical replicates each. Quantitative data are presented as a mean ± S.E.M. The representative images were acquired by SRRF imaging. Two-sided unpaired student's *t* test: Figs. 3a, b, d, e. Two-sided one-way ANOVA: Fig. 3c. Two-sided two-way ANOVA: Figs. 3f–m. ** indicates $p < 0.01$ versus control. ## indicates $p < 0.01$ versus corticosterone in hippocampal neurons and cortisol in SH-SY5Y. Data are provided as a Source data file.

hippocampus finally culminates in neurodegeneration and behavior change such as spatial memory and major depressive disorder. Y-maze test evaluates the spatial memory to use the innate nature of rodents to explore the new objects. We observed that corticosterone-treated mice exhibited impaired spatial memory, but mice pretreated with PMA demonstrated attenuated damaged cognitive function (Fig. 8e). The forced swim test evaluates the depressive-like behavior in rodents. Depressed mice tend to remain immobile in the cylinder, indicating a lack of willingness to escape their environment owing to depressive mood. There were no significant changes in time to mobility between the experimental groups, meaning that depressive-like behavior was not observed by corticosterone (Fig. 8f). We also confirmed that pretreatment with the GR antagonist RU 486 rescued the hippocampus from inhibitory effect of corticosterone on PGC1α and NIX expression (Fig. 8g). To confirm that only stress-induced levels of corticosterone trigger mitophagy inhibition, synaptic dysfunction, and spatial memory deficits, dose-dependent effects of corticosterone were also determined. We observed that only high concentrations of corticosterone (10 mg/kg) decreased mitophagy levels and synaptic markers (Supplementary Fig. 5b). We also showed that 10 mg/kg corticosterone decreased spatial memory while vehicle treatment or 1 mg/kg corticosterone treatment groups showed normal memory function (Supplementary Fig. 5c). Altogether, NIX-dependent mitophagy and spatial memory are inhibited under high levels of corticosterone via activation of the GR-PGC1α pathway in vivo.

## Discussion

Our study provides evidence that chronic and high-dose glucocorticoid exposure damages NIX-dependent mitophagy and subsequent synaptic homeostasis, but restoration of NIX levels reverses this effect both in vivo and in vitro (Fig. 8h). Glucocorticoids have both good and bad effects on synapses. Acute stress (during a few hours) can enhance synaptic function, but chronic stress contributes to synaptic damage by changing their structure and metabolism[35]. Neurons exposed to glucocorticoids first adapt to the stress state by releasing different proportions of neurotransmitters which enhance synaptic transmission to aid in increasing cellular energy; however, consistent glucocorticoid treatment results in infrequent long-term potentiation and reducing synaptic plasticity due to excessive glutamate release[36,37]. We assume that glucocorticoids would, in the first instance, enhance mitophagy to maintain mitochondria quality because releasing neurotransmitters is very energy-demanding; however, the differing responses to acute and chronic exposure to glucocorticoids requires further study. Furthermore,

we showed that the long-term exposure of neurons to stress-induced levels of corticosterone induced perinuclear accumulation of mitochondria, which failed to undergo mitophagy. We previously demonstrated that high levels of glucocorticoids reduced anterograde trafficking of mitochondria via SCG10-mediated microtubule destabilization[5]. Glucocorticoids also induce hyperphosphorylation of tau and inhibit microtubule assembly, culminating in inhibition of microtubule-dependent intracellular trafficking[1,38]. It is possible that the microtubule destabilizing effect of glucocorticoids also affect the perinuclear clumping of mitochondria as retrograde transport may also be impaired. However, we observed that the damaged mitochondria accumulated in somatodendritic compartment due to this inhibition were not eliminated through mitophagy, consistent with previous studies[39,40]. From these observations, we suggest that glucocorticoids decrease mitophagy and inhibit the anterograde transport. Furthermore, our data raised the need to identify what level of glucocorticoids is required to alter mitophagy. Troncoso et al. demonstrated that autophagy stimulation by dexamethasone induces mitochondrial clearance in muscle tissues[41]. The purpose of another study was to improve cell survival by increasing autophagy because glucocorticoids are a common drug used to alleviate inflammation in muscles and bones. In that study, the authors used 10-fold less glucocorticoids to treat the cells[42]. In contrast, glucocorticoids triggered defective mitophagy in this study. Among the various neuronal tissues, hippocampal cells are the most sensitive cell type to stress-induced glucocorticoid exposure. A recent article also suggested that autophagy activation significantly prevents glucocorticoid-increased adiposity, indicating that maintaining cellular homeostasis via autophagy is difficult in the presence of excessive amounts of glucocorticoids[43]. Collectively, with regard to the effect of high levels of glucocorticoids on mitochondria and autophagy, it is reasonable to conclude that the stressed state can strongly and negatively affect mitophagy and synaptic homeostasis.

We then determined that a distinct mechanism how stress-induced glucocorticoids oppose the process of mitophagy. The PINK1-parkin-mediated mitophagy has a protective effect for patients with neurodegenerative disease[8]. However, current studies reported that the PINK1-parkin pathway is well explained in stress/toxin-induced mitophagy but not in physiological mitophagy, also known as basal mitophagy which continuously occurs to eliminate effete mitochondria. Receptor-mediated mitophagy regulators, mainly responsible for basal mitophagy, can compensate for or maintain a distinct homeostatic mechanism away from PINK1-parkin-mediated mitophagy in a case of cerebral ischemic injury, PD, and AD[14,44,45]. Interestingly, we also showed

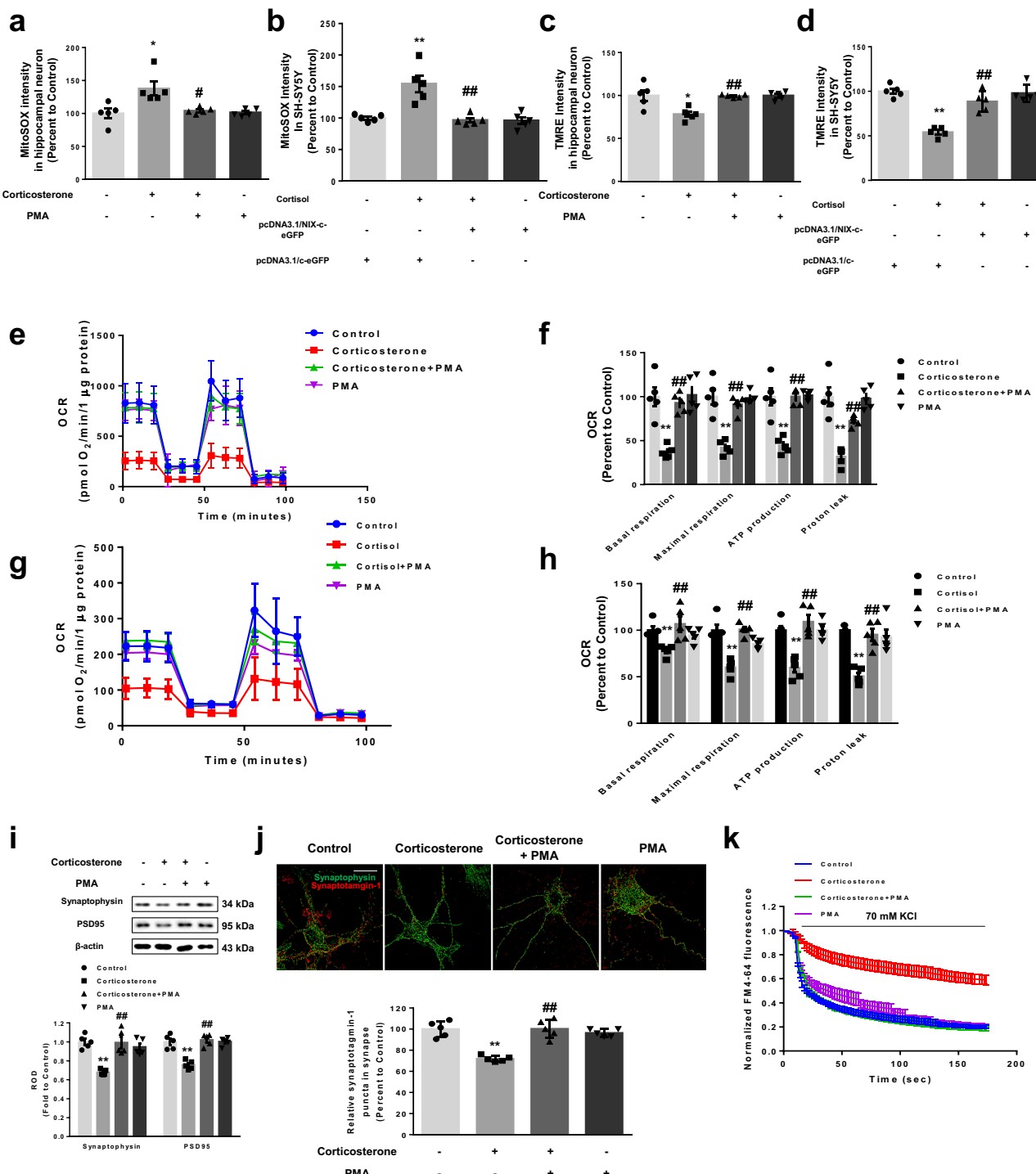

that glucocorticoids selectively repress NIX expression, which is strongly affected by stress according to previous report[15]. Knocking out of *PARK2* is enough compensated for by the activation of receptor-mediated mitophagy in neurons, suggesting that basal mitophagy may be dominant in neural tissue[46]. In agreement of this view, previous research insisted that basal mitophagy is especially enhanced in several tissues including brain, independent from PINK1-parkin pathway[12]. Based on previous results, we may infer from our results that glucocorticoids suppress NIX-mediated basal mitophagy and are not associated with PINK1-parkin-mediated mitophagy potential uncoupling in neuronal cells owing to the lack of responsiveness to exogenous stressors such as antimycin A1. There is another

controversy regarding the relationship between NIX and parkin. Some research suggests that NIX recruits parkin to maintain mitochondrial homeostasis or parkin activates NIX via the regulation of serine 81 phosphorylation[47]. Conversely, parkin is thought to be independent from NIX-induced basal mitophagy and only associated with stressor/toxin-mediated mitophagy. Although further determination of the molecular relationship between NIX and parkin is necessary, our results indicate that NIX expression remains unchanged with parkin overexpression. Furthermore, we investigated whether *PARK2* knockdown can stimulate NIX-dependent mitophagy under glucocorticoid treatment but it turned out that the compensation offered by reduction in PINK1-parkin pathway appears to be insignificant.

**Fig. 4 NIX upregulation recovers glucocorticoid-induced mitochondrial dysfunction. a, b** Hippocampal neurons were pretreated with phorbol 12-myristate 13-acetate (PMA, 10 nM) for 30 min prior to corticosterone for 24 h. SH-SY5Y cells were transfected with pcDNA3.1/c-eGFP or pcDNA3.1/NIX-c-eGFP vector for 24 h prior to cortisol treatment for 24 h. The level of mtROS was measured via MitoSOX staining with luminometer. $n = 5$. **c–d** Hippocampal neurons were pretreated with PMA (10 nM) for 30 min prior to corticosterone for 48 h. SH-SY5Y cells were transfected with pcDNA3.1/c-eGFP or pcDNA3.1/NIX-c-eGFP vector for 24 h prior to cortisol treatment for 48 h. Mitochondrial membrane potential was measured via tetramethylrhodamine ethyl ester (TMRE) staining with luminometer. $n = 5$. **e–h** Both hippocampal neurons and SH-SY5Y cells were pretreated with PMA for 30 min before corticosterone and cortisol treatment for 48 h, respectively. Oxygen consumption rate (OCR) changes under mitochondrial stress test were measured by using Seahorse SF24 Extracelluar Flux analyzer where oligomycin, carbonyl cyanide-4-(trifluoromethoxy)phenylhydrazone (FCCP), and antimycin A/rotenone mixture were treated. $n = 5$. Statistics of basal respiration, maximal respiration, ATP production, and proton leak were also presented. **i–k** Hippocampal neurons were pretreated with PMA for 30 min prior to corticosterone for 48 h. **i** Synaptophysin and PSD95 were detected by western blot. Loading control is β-actin. $n = 5$. **j** After stimulation with depolarization buffer, cells were incubated with synaptotagmin-1 antibody. Synaptophysin (green) was used to visualize synapse and synaptotagmin-1 (red) was used for measuring uptake of synaptic vesicles. Scale bars, 20 μm (magnification, ×1000). $n = 5$. **k** Conditioned hippocampal neurons were stained with FM4-64 dye and stimulated with high $K^+$ buffer for destaining. Time-lapse imaging was performed over 180 sec at 1 sec intervals with an Eclipse Ts2™ fluorescence microscopy. $n = 5$. All blots and immunofluorescence images are representative. $n = 5$ from independent experiments with two technical replicates each. Quantitative data are presented as a mean ± S.E.M. The representative images were acquired by SRRF imaging system. Two-sided two-way ANOVA was conducted. *, ** indicates $p < 0.05$, $p < 0.01$ versus control, respectively. #, ## indicates $p < 0.05$, $p < 0.01$ versus corticosterone in hippocampal neurons and cortisol in SH-SY5Y, respectively. Data are provided as a Source data file.

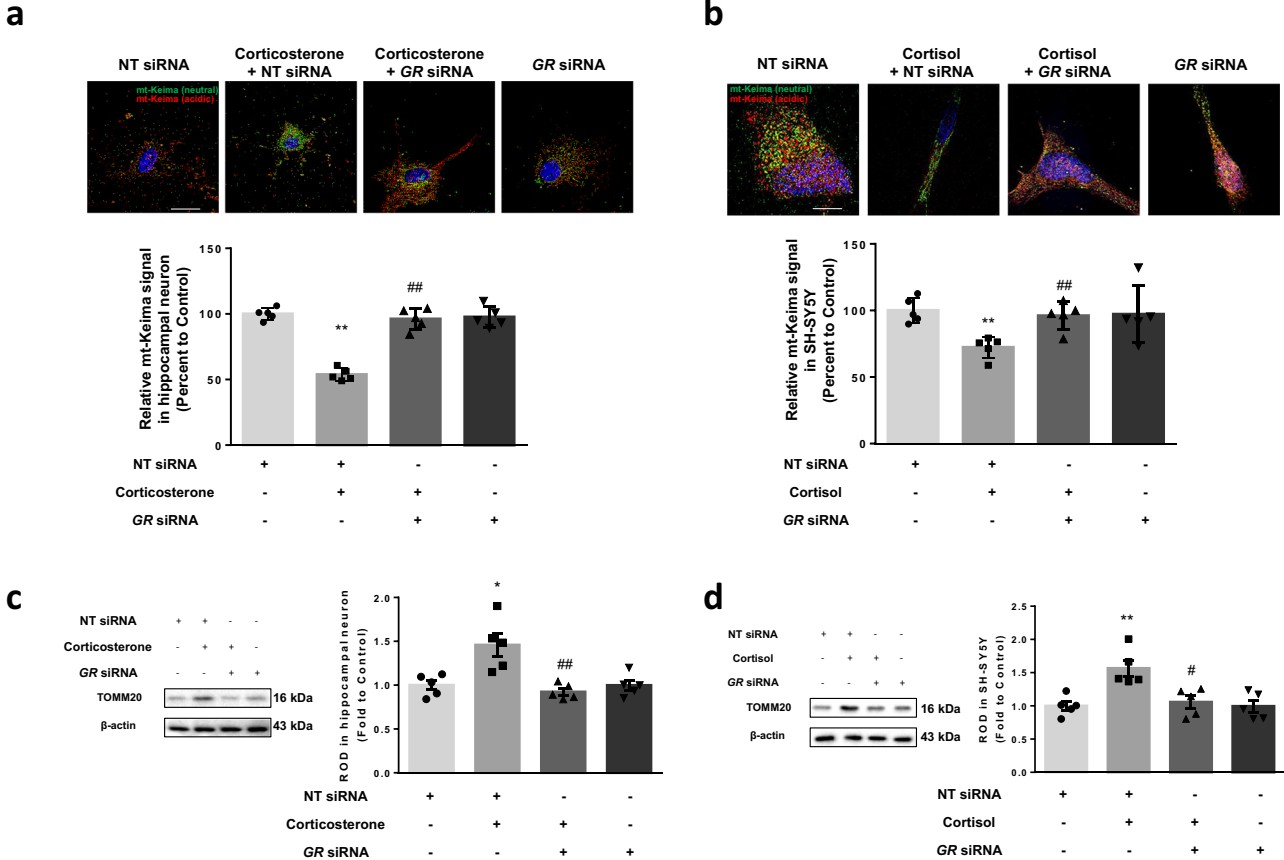

**Fig. 5 GR-dependent inhibition of mitophagy. a–d** Nontargeting (NT) or *GR* siRNA was transfected to hippocampal neurons and SH-SY5Y cells for 24 h prior to corticosterone and cortisol for 24 h, respectively. **a, b** Hippocampal neurons were transfected with mitochondria-targeted Keima (mt-Keima) at DIV 5, whereas SH-SY5Y cells were transfected with mt-Keima 48 h prior to treatment. Green and red fluorescence of mt-Keima were visualized. DAPI was used for nuclear counterstaining (blue). Ratio of red to sum of red and green was quantified. Scale bars, 20 μm (magnification, ×1000). $n = 5$. **c–d** TOMM20 levels were detected by western blot. Loading control is β-actin. $n = 5$. All blots and immunofluorescence images are representative. $n = 5$ from independent experiments with two technical replicates each, respectively. Quantitative data are presented as a mean ± S.E.M. The representative images were acquired by SRRF imaging system. Two-sided two-way ANOVA was conducted. *, ** indicates $p < 0.05$, $p < 0.01$ versus control, respectively. #, ## indicates $p < 0.05$, $p < 0.01$ versus corticosterone in hippocampal neurons and cortisol in SH-SY5Y, respectively. Data are provided as a Source data file.

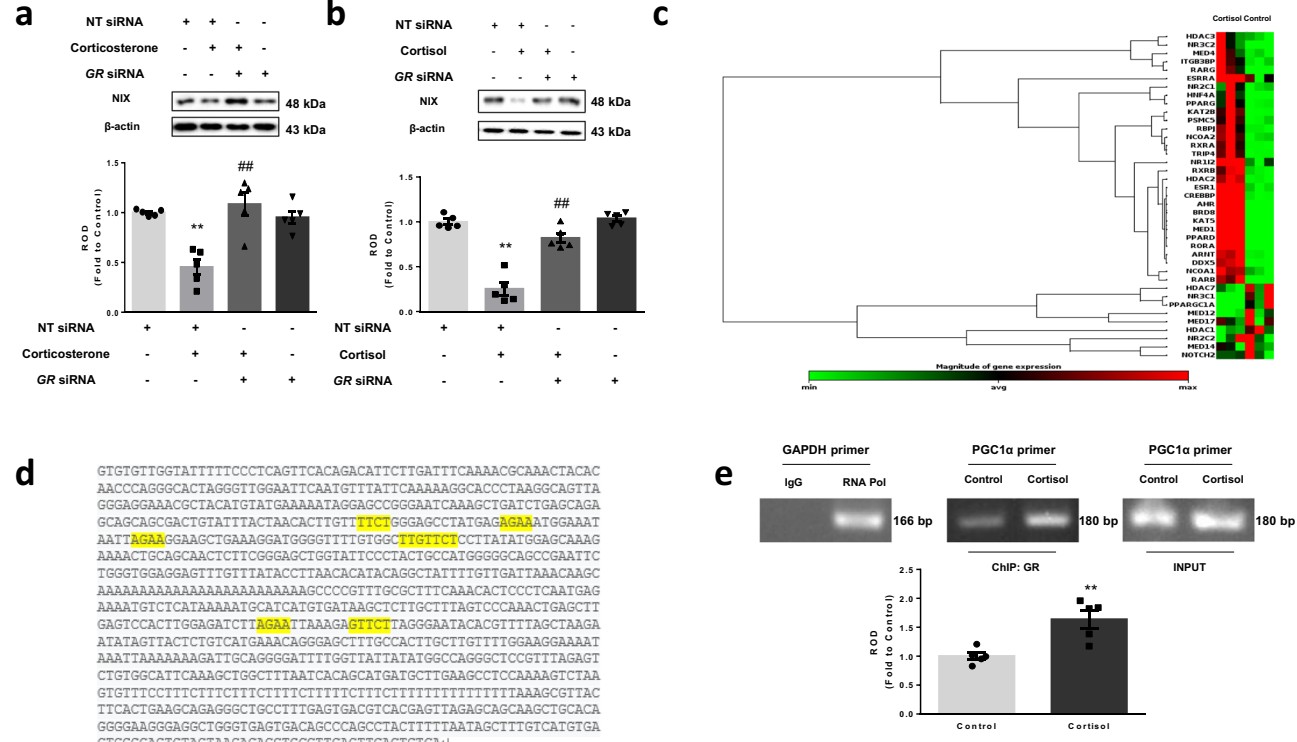

**Fig. 6 GR-dependent downregulation of NIX expression via PGC1α. a**, **b** Nontargeting (NT) or *GR* siRNA was transfected to hippocampal neurons and SH-SY5Y cells for 24 h prior to corticosterone and cortisol for 24 h, respectively. NIX expression was detected in western blot where β-actin was used as a loading control. *n* = 5. **c** SH-SY5Y cells were incubated with cortisol for 6 h. The mRNA expression levels of genes associated with nuclear receptors and coregulators were assessed by RT2 Profiler PCR array. Heat maps with hierarchical clustering were acquired by using the GeneGlobe Data analysis Center on Qiagen website. *n* = 3. **d** A thousand base pair upstream of the first codon of the *PPARGC1A* was described and the putative GRE binding sequence was emphasized with yellow labeling. **e** SH-SY5Y cells were incubated with cortisol for 6 h. DNA was immunoprecipitated with IgG, RNA polymerase (RNAPol), and glucocorticoid receptor (GR) antibody. The immunoprecipitation and input samples were amplified with primers of *GAPDH* and *PPARGC1A* gene. *n* = 5. All blots are representative. *n* = 3 or 5 from independent experiments with two technical replicates each, respectively. Quantitative data are presented as a mean ± S.E.M. Two-sided two-way ANOVA was conducted in Fig. 6a, b. Two-sided unpaired student's *t* test was conducted in Fig. 6e. ** indicates *p* < 0.01 versus control. ## indicates *p* < 0.01 versus corticosterone in hippocampal neurons and cortisol in SH-SY5Y, respectively. Data are provided as a Source data file.

We would refute the paradigm that PINK1-parkin is critical for NIX-dependent mitophagy, because NIX alone can maintain basal mitophagy and compensate for other mitophagy regulators, demonstrated in the cases of mouse and primates[10]. The next question asks why glucocorticoids primarily affect basal mitophagy in neural tissue. The characteristics of neural tissues are very different from those of other tissues. It is well-known that mitochondria uncoupling factors, such as carbonyl cyanide-4-(trifluoromethoxy)phenylhydrazone (FCCP) are not very applicable to neural cells. Furthermore, many reports suggest that basal mitophagy is more relevant in pathological conditions of neural tissue, because the post-mitotic cell immediately requires degradation of dysfunctional mitochondria following damage[48,49]. Because basal mitophagy occurs constitutively in brain, chronic exposure of high-dose glucocorticoids that adversely affect mitochondrial function is more likely to suppress basal mitophagy than PINK1 in neurons. However, mitochondrial membrane potential changes through subsequent glucocorticoid treatment may later affect the PINK1-parkin pathway during mitochondrial dysfunction. Studies using other experimental conditions or long-time observations will, therefore, be required. Our findings provide evidence that glucocorticoids are mostly responsible for inhibiting NIX-dependent mitophagy but do not evoke the PINK1-parkin mitophagy pathway, which does not compensate for the reduced level of basal mitophagy.

That NIX alone can play a neuroprotective role in the glucocorticoid-induced impairment of mitophagy and synaptic dysfunction is an important finding. However, NIX is also known to be a BH3-only proapoptotic protein inducing cell death. More recent work has demonstrated that the BH3 domain in NIX is weakly conserved and redundant for proapoptotic function unlike typical BH3-only proteins[50]. Furthermore, our results join the growing evidence that suggests that clearance of damaged mitochondria by NIX-mediated mitophagy prevents the accumulation of dysfunctional mitochondria, resulting in increased healthy mitochondria[15]. Therefore, many brain-related diseases correlate with NIX deficiency. Brain damage was more serious in *Nix* knockout mice, whereas more neurons survived following NIX upregulation after cerebral ischemia-reperfusion injury or traumatic brain injury[15,51,52]. Furthermore, NIX-mediated mitophagy evokes autophagic cell survival in glioblastoma[53]. Taken together, we concluded that dysregulation of mitophagy in neuron due to NIX deficiency by stress-induced levels of glucocorticoids leads to the accumulation of damaged mitochondria, eventually contributes to synaptic dysfunction and cell death.

The GR-mediated pathway is only activated by elevated levels of glucocorticoids[35]. Among the many GR-mediated pathways, we found that PGC1α is a strong mediator between nuclear GR and NIX according to PCR array and in silico assay results. Despite the existence of transcription factors such as Sp1, FoxO3, and Hif1α that upregulate NIX expression and are induced by

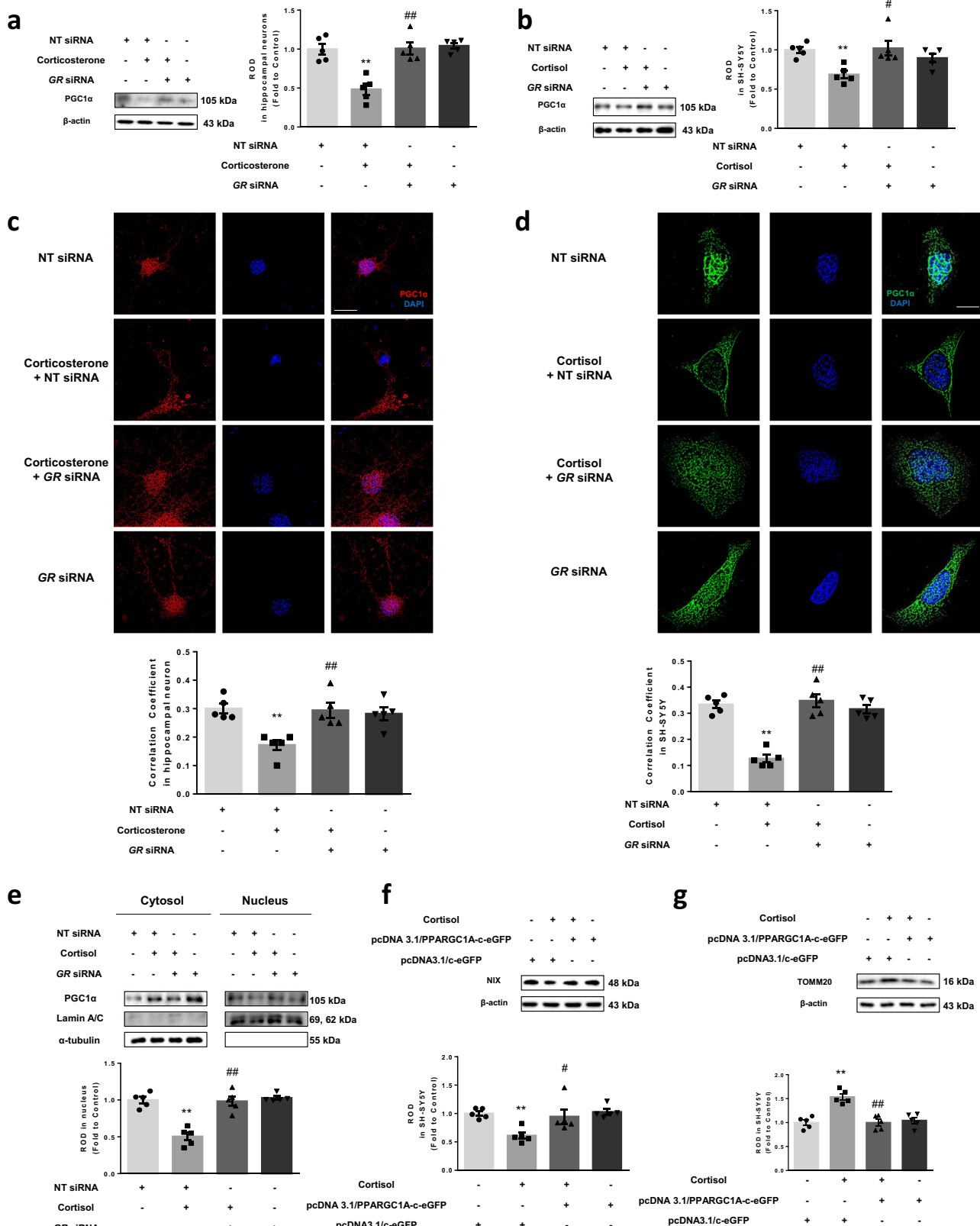

ROS, our findings suggest that GR in particular trans-represses the PGC1α gene, which has many GRE sequences in its promoter. It was also demonstrated that GR is highly associated with PPARγ-dependent signaling[54]. The effect of corticosterone and cortisol-induced ROS on receptor-mediated mitophagy may not be negligible, but recent studies suggest that NIX regulation does not depend on the hypoxic state in various tissues[50,55]. Because decreased PGC1α eventually causes an increase in mtROS, which subsequently may induce other types of receptor-mediated or NIX-dependent mitophagy, the regulation of mitophagy by glucocorticoids needs to be further investigated. However, our results suggest that NIX is significantly inhibited by GR-dependent

**Fig. 7 Role of PGC1α in NIX-dependent mitophagy. a–e** Nontargeting (NT) or *GR* siRNA was transfected to hippocampal neurons and SH-SY5Y cells for 24 h prior to corticosterone and cortisol for 12 h, respectively. **a, b** Peroxisome proliferator-activated receptor gamma coactivator 1-alpha (PGC1α) expression was detected in western blot where β-actin was used as a loading control in both cell types. n = 5. **c** Colocalization of PGC1α (red) and DAPI (blue) in hippocampal neurons was visualized with SRRF imaging system. Scale bars, 20 μm (magnification, ×1000). n = 5. **d** Colocalization of PGC1α (green) and DAPI (blue) in SH-SY5Y was visualized with SRRF imaging system. Scale bars, 20 μm (magnification, ×1000). n = 5. **e** PGC1α protein expressions in subcellular fraction samples were detected by western blotting. Lamin A/C and α-tubulin were used as a nuclear and cytosolic loading control, respectively. n = 5. **f, g** SH-SY5Y cells were transfected with pcDNA3.1/c-eGFP or pcDNA3.1/PPARGC1A-c-eGFP vector for 24 h prior to cortisol treatment for 24 h. **f** NIX expression was detected in western blot where β-actin was used as a loading control. n = 5. **g** TOMM20 levels were detected by western blot. Loading control for western blot is β-actin. n = 5. All blots and immunofluorescence images are representative. n = 5 from independent experiments with two technical replicates each. Quantitative data are presented as a mean ± S.E.M. Two-sided two-way ANOVA was conducted. ** indicates p < 0.01 versus control. #, ## indicates p < 0.05, p < 0.01 versus corticosterone in hippocampal neurons and cortisol in SH-SY5Y, respectively. Data are provided as a Source data file.

pathway. Because GR is most abundant and NIX is second abundant in hippocampus throughout the brain regions according to RNA sequencing data from the mouse brain in the Allen Brain Atlas, GR-induced NIX suppression is likely to dominant over other ROS-related signaling pathways involving glucocorticoids in the hippocampus. In addition, it has been reported that the strong correlation between PGC1α and receptor-mediated mitophagy contributes to mitophagy activation[10,56,57]. Furthermore, downregulation of PGC1α in brain followed by decreased synaptic density has been observed in many neurodegenerative disorders including AD and Parkinson's disease[34,58]. Consistent with these previous research, we suggest that PGC1α transcription decreased by glucocorticoids selectively downregulates NIX proteins. Our results partially agree with a recent report, suggesting that PGC1α is a strong activator of BNIP3-mediated mitophagy in the dopaminergic neurons of rat midbrain[44]. However, we showed that basal mitophagy activated by the PGC1α-NIX axis is more strongly and selectively affected by glucocorticoids in hippocampal and neuroblastoma cells. Collectively, our results suggest that mitophagy through the GR-PGC1α-NIX axis is dominant in hippocampal cells and susceptible to the glucocorticoids due to the large number of GRs.

Our in vivo data also suggest that the reduction in NIX-dependent mitophagy by glucocorticoids induces progression of synaptic abnormality and impairment of spatial memory, but not affecting depressive-like behavior. Several reports indicated that 7 days of glucocorticoid treatment is not enough to trigger depression-like behavior in mice or rats, whereas 14 days of treatment triggered these behaviors[59]. The onset time of clinical symptoms of spatial memory and mood disorder mainly regulated by dorsal and ventral hippocampus can be explained owing to the different responsiveness of both regions[20]. Thus, prolonged exposure of corticosterone can evoke depression in our in vivo models. Indirect effects of glucocorticoids can also affect neuronal mitophagy in vivo. For example, gluconeogenesis by glucocorticoids increase blood glucose levels, which indirectly decrease mitochondrial motility and inhibit mitophagy[60]. However, GRs are most abundant in hippocampus, contributing to its glucocorticoid responsiveness. Thus, hippocampus may represent the primary site of mitophagy inhibition as GR-PGC1α-NIX axis is actively triggered[61]. Furthermore, alterations in complexed connectivity between glia and neurons by glucocorticoid signaling could affect mitophagy levels in neurons via neuroregulatory molecules released from activated astrocytes or microglia. Our in vivo results suggest that mitophagy can be inhibited following downregulation of NIX in glia by glucocorticoids; however, we focused more on neuronal mitophagy. Unlike non-neuronal cell types, the neurodegenerative effect is maximized by neurons, which are more sensitive to abnormal mitophagy owing to their post-mitotic nature, tremendous ATP demand, and extraordinary cellular shapes. Furthermore, behavior changes such as memory

dysfunction and mood disorder are the representative phenotypes of hippocampal neurodegeneration. However, further investigation of indirect effects of glucocorticoids on mitophgy is necessary to learn more complexly and deeply about the systemic effects in in vivo. In conclusion, our study presents the identification of NIX as a potential target protein for treating glucocorticoid-induced synaptic defects and spatial memory impairments at an early stage by improving mitophagy.

## Methods

**Materials.** Fetal bovine serum (#SH30088.031 R) and antibiotic–antimycotic mixture (#15240062) were purchased from Hyclone (Logan, UT, USA) and Gibco (Grand Island, NY, USA), respectively. The antibodies of BNIP3 (#sc-56167, 1:1000 for western blot), Lamin A/C (#sc-20681, 1:1000 for western blot), and β-actin (#sc-47778, 1:1000 for western blot) were obtained from Santa Cruz Biotechnology (Paso Robles, CA, USA). The antibodies of TOMM20 (#ab56783, 1:3000 for western blot, 1:500 for immunostaining), synaptophysin (#ab32127, 1:3000 for western blot, 1:500 for immunostaining), γ-synuclein (#ab55424, 1:500 for immunostaining), MAP2 (#ab11267, 1:500 for immunostaining), LAMP1 (#ab24170, 1:500 for immunostaining), and parkin (#ab77924, 1:3000 for western blot) were purchased from Abcam (Cambridge, MA, USA). The GR antibody (#12041S, 1:3000 for western blot, 1:500 for immunostaining) was obtained from Cell Signaling Technology, Inc. (Danvers, MA, USA). The antibodies of LC3 (#NB100-2220, 1:3000 for western blot), PGC1α (#NBP1-04676, 1:3000 for western blot, 1:200 for immunostaining), NIX (#NBP1-88558, 1:3000 for western blot, 1:200 for immunostaining), and PINK1 (#BC100-494, 1:3000 for western blot) were purchased from Novus Biologicals (Littleton, CO, USA). The antibodies of synaptotagmin-1 (#PA5-27935, 1:100 for synaptotagmin uptake assay), Tau5 (#AHB0042, 1:200 for immunostaining), and ubiquitin (#701339, 1:1000 for western blot, 1:100 for immunostaining) were obtained from Thermo Fisher (Rockford, IL, USA). Mfn1 antibody (#66776-1-lg, 1:100 for immunostaining) was acquired from Proteintech (Rosemont, IL, USA). Cortisol (#H4001), corticosterone (#C2505), RU 486 (#M8046), DAPI (#23397), bafilomycin A1 (#B1793), antimycin A1 (#A0149), CCCP (#C2759), PMA (#P8139), PSD95 antibody (#MAB1596, 1:1000 for western blot, 1:500 for immunostaining), and α-tubulin antibody (#T6074, 1:5000 for western blot) were purchased from Sigma Chemical Company (St. Louis, MO, USA). The plasmids for pcDNA3.1/NIX-c-eGFP, pcDNA3.1/PPARGC1A-c-eGFP, and pcDNA3.1/c-eGFP were purchased from KomaBiotech (Seoul, Korea). DsRed2-mito (#632421) and AutophagSense (#632583) were obtained from Takara (Otsu, Shinga, Japan). Vector for mt-Keima (#AM-V0251) and Keima-Red-Parkin (#AM-V0259M) were purchased from MBL (Nagoya, Japan).

**Cell culture.** Cultures of mouse hippocampal neurons were performed as described following modified protocol[62]. Hippocampal neurons from E18 mouse embryos are used in compliance and approval with the Institutional Animal Care and Use Committee of Seoul National University (SNU-190523-1-1). In brief, hippocampal neurons were cultured at low density on coverslips with poly-D-lysine inverted over a feeder layer of astrocytes or high density on poly-D-lysine coated six-well plates in neurobasal medium (Gibco) supplemented with 2% B27 supplement (Gibco). The human neuroblastoma cell line SH-SY5Y was acquired by Korean Cell Line Bank (Seoul, Korea). SH-SY5Y cells were cultured in high-glucose Dulbecco's Modified Eagle Medium (DMEM, Hyclone) containing 10% fetal bovine serum and 1% antibiotic–antimycotic mixture kept at 37 °C with 5% $CO_2$. For serum reduction, SH-SY5Y cells were cultured with serum-free DMEM containing 1% antibiotic–antimycotic solution for 24 h.

**Transfection of plasmid DNA.** Prior to treatment, cells were incubated with a mixture of plasmid DNA, MEM, and lipofectamine 3000 (Thermo Fisher).

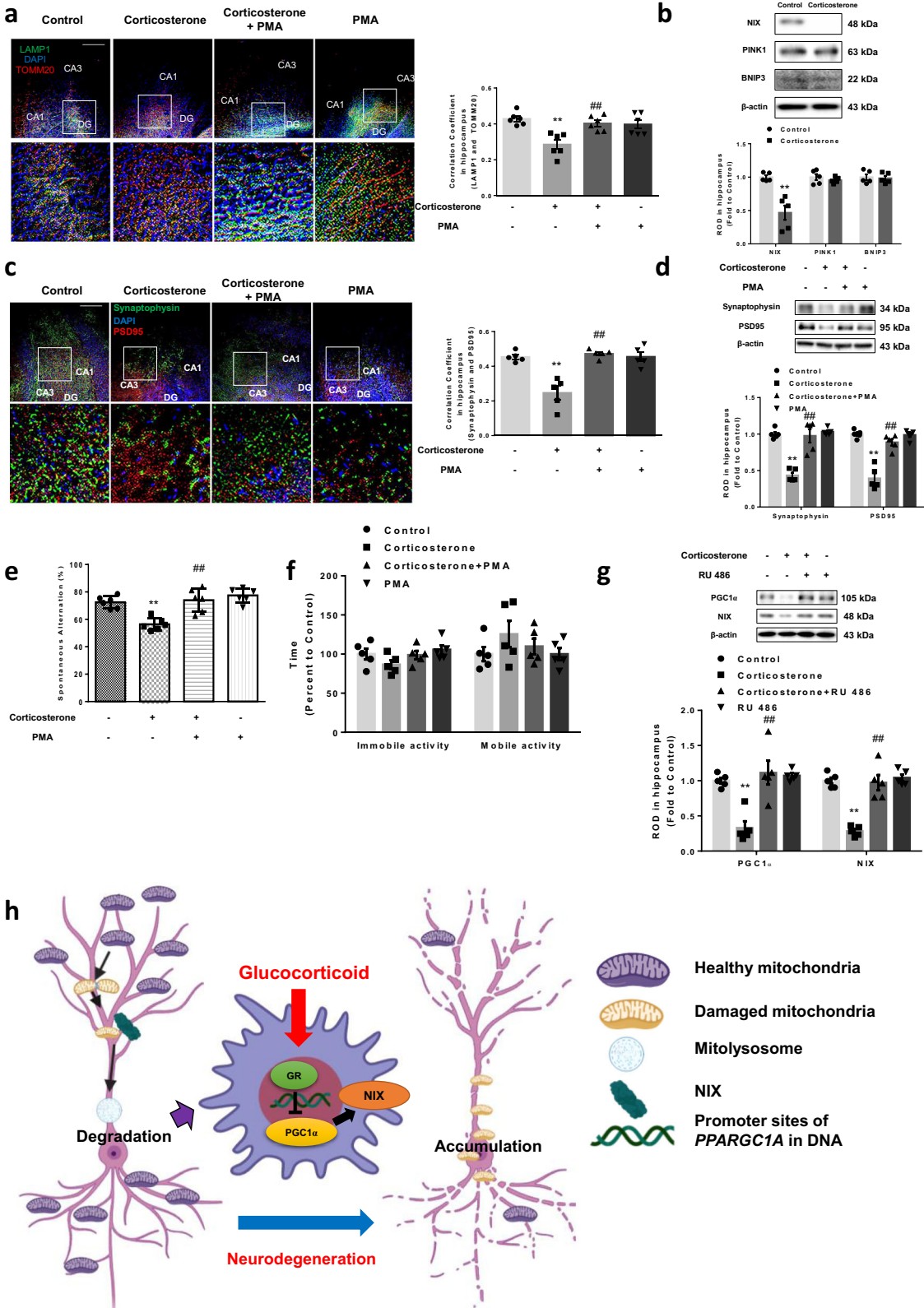

Lipid-based transfection method with lipofectamine 3000 was used in neuron coverslips into a 12-well plate following modified protocol[62]. After 60–90 min, neurons were replaced with fresh neurobasal media containing 2% B27 supplement. SH-SY5Y cells were transfected with plasmids using lipofectamine 3000 following modified protocol[63]. After incubation for 6 h, lipofectamine reagent was removed and replaced with fresh starvation media.

**Transfection of siRNA for gene silencing.** Cells were grown until 60% confluency of the plate. Prior to corticosterone and cortisol treatment, cells were incubated with 25 nM of the indicated siRNAs and turbofect transfection reagent (Thermo Fisher) for 24 h according to the manufacturer's instructions. The siRNAs specific for human/mouse *GR* are purchased from Bioneer Corporation (Daejeon, Korea). The siRNAs specific for human *BNIP3L* and nontargeting (NT) are obtained from Dharmacon (Lafayette, CO, USA).

**Fig. 8 Corticosterone affects NIX-dependent mitophagy through decreasing PGC1α in vivo. a–f** Mice were exposed to vehicle, corticosterone (10 mg/kg), corticosterone with phorbol 12-myristate 13-acetate (PMA pretreatment, 200 μg/kg), or PMA alone for 7 days. **a** Slide samples for IHC were immunostained with LAMP1 (green), TOMM20 (red), and DAPI (blue). Scale bars, 100 μm (magnification, ×200). n = 5. **b** The expressions of NIX, PTEN-induced kinase 1 (PINK1), and BCL2 interacting protein 3 (BNIP3) were detected with western blot where β-actin was used as a loading control. n = 5. **c** Slide samples for IHC were immunostained with synpatophysin (green), PSD95 (red), and DAPI (blue). Scale bars, 100 μm (magnification, ×200). n = 5. **d** Synaptophysin and PSD95 were detected by western blot. Loading control is β-actin. n = 5. **e** The mice were subjected to Y-maze test to evaluate spatial memory function. n = 6. **f** The mice were subjected to forced swim test to evaluate depression-like behavior. n = 5. **g** Vehicle or RU 486 (5 mg/kg) injected mice were presented with/without corticosterone (10 mg/kg) for 3 days. The expressions of peroxisome proliferator-activated receptor gamma coactivator 1-alpha (PGC1α) and NIX were visualized via western blotting. Loading control is β-actin. n = 5. **h** The schematic model for mechanisms of inhibition in NIX-dependent mitophagy by glucocorticoid was shown. All blots and immunofluorescence images are representative. n = 5 or 6 from each animal with two technical replicates each in results of IHC and western blot. Quantitative data are presented as a mean ± S.E.M. The representative images were acquired by SRRF imaging system. Two-sided two-way ANOVA was conducted except Fig. 8b, data of which were analyzed by two-sided unpaired student's t test. ** indicates p < 0.01 versus control and ## indicates p < 0.01 versus corticosterone, respectively. Data are provided as a Source data file.

**Reverse transcription-PCR and real-time PCR.** RNA samples were extracted using MiniBEST Universal RNA extraction kit (Takara). Reverse transcription was performed using 1 μg of RNA for 1 h at 45 °C followed by 5 min at 95 °C with a Maxime RT-PCR premix kit (Intron Biotechnology, Seongnam, Korea) to obtain cDNA. The cDNA samples were then amplified using Quanti NOVA SYBR Green PCR Kits (Qiagen, Hilden, Germany). Real-time quantification of RNA targets was performed in a Rotor-Gene 6000 real-time thermal cycling system (Corbett Research, NSW, Australia). Quantitative real-time PCR was performed as follows: 10 min at 95 °C for DNA polymerase activation; 50 cycles of 15 sec at 94 °C, 30 sec at 54 °C, and 30 sec at 72 °C. Data collected during the extension step, and melting curve analysis was validated to verify the specificity and identity of the PCR products. Normalization of gene expression levels was performed by using the *ACTB* gene as a control.

**RT2 human nuclear receptors and coregulators PCR array.** The RT2 profiler PCR (#PAHS-056Y, Qiagen) was used to analyze gene expression of human nuclear receptors and coregulators in cells treated with cortisol for 6 h according to the manufacturer's instructions. In this array, a set of optimized primer assays allows mRNA transcript detection for 84 genes and five housekeeping genes in 96 wells detected by StepOnePlus real-time PCR system (Thermo Fisher). PCR array data were analyzed using the GeneGlobe Data Analysis Center on Qiagen's website. The genes with p value below 0.05 were selected.

**DNA extraction and mtDNA content (damage) assay.** DNA samples were extracted using AccuPrep® Genomic DNA extraction kit (Bioneer Corporation) according to the manufacturer's instruction. Both mtDNA and nDNA levels in the same experimental condition were evaluated by real-time PCR described above. The real-time PCR using mtDNA primer targeted to mitochondrial *16SrRNA* gene and nDNA primer targeted to *B2m/β2*-microglobulin gene is a well-known assessment of mitochondrial contents. The mtDNA normalized to nDNA was quantified for assessing amount of mtDNA content[27]. Primer sequences were obtained from previous study[45].

**Western blot analysis.** Harvested samples were incubated with the appropriate buffer containing protease and phosphatase inhibitors for 30 min on ice. Cell debris was cleared by centrifugation (13,000 × g at 4 °C for 30 min). Protein determination was performed by a bicinchoninic acid quantification assay kit (Thermo Fisher). Equal amounts of sample proteins were prepared for 8–15% sodium dodecyl sulfate polyacrylamide gel electrophoresis and then transferred to a polyvinylidene fluoride membrane. The membranes were blocked with 5% bovine serum albumin (Sigma Chemical Company) in tris-buffered saline containing 0.2% Tween-20 (TBST; 150 mM NaCl, 10 mM Tris-HCl (pH 7.6), 0.1% Tween-20) solution for 30 min. Blocked membranes were incubated with primary antibodies at 4 °C overnight. The membranes were washed three times and incubated with the horseradish peroxidase-conjugated secondary antibodies (Thermo Fisher, 1:10,000) at room temperature for 2 h. Western blotting bands were detected with chemiluminescence solution (BioRad, Hercules, CA, USA) and the densitometry analysis for quantification was carried out by using Image J 1.53 software (developed by Wayne Rasband, National Institutes of Health, Bethesda, MD, USA).

**Co-immunoprecipitation.** Primary antibodies were immobilized with SureBeads protein G magnetic beads (BioRad). Immobilized magnetic beads were incubated with the total lysates of cells (300 μg) at 4 °C overnight. Magnetic beads were pulled-down by magnet and then collected. The antibody-bound protein was acquired by incubation in elution buffer (Thermo Fisher). Protein analysis was performed by western blot where anti-mouse or rabbit IgG antibodies were used as a negative control.

**Immunocytochemistry.** Hippocampal neurons on a coverslip or SH-SY5Y cells on a confocal dish (Thermo Fisher) were fixed with 4% paraformaldehyde for 15 min and incubated in 0.3% Tween-20 for 5 min. Cells were placed in 5% normal goat serum (NGS) in PBS for 1 h and incubated with primary antibody dissolved in 5% NGS for overnight in 4 °C. Next, the cells were incubated for 2 h at room temperature with Alexa Fluor™ secondary antibodies (Thermo Fisher, 1:100). Images were acquired by super-resolution radial fluctuations (SRRF) imaging system (Andor Technology, Belfast, UK). The fluorescent intensity analysis and colocalization analysis with Pearson's correlation coefficient were acquired by Fiji 1.53 software (developed by Wayne Rasband, National Institutes of Health, Bethesda, MD, USA). The values of Pearson's correlation coefficient or the relative ratio of Pearson's correlation coefficient (Compared to control) were shown in the quantification graph next to immunofluorescence images.

**Measurements of mtROS and mitochondrial membrane potential.** MitoSOX Red (#M36008, Thermo Fisher) and TMRE (#87917, Sigma Chemical Company) were used for measuring mtROS and mitochondrial membrane potential, respectively. The concentrations were decided following the manufacturer's instructions. The fluorescence intensity was measured at luminometer (Victor3; Beckman Coulter, Fullerton, CA, USA) and normalized to total cell count.

**Measurement of mitochondrial biogenesis using MitoBiogenesis ™ In-Cell ELISA kit.** MitoBiogenesis™ In-Cell ELISA kit purchased from abcam (#ab140359) for measuring mitochondrial biogenesis was used in SH-SY5Y cells. Two proteins are detected following the manufacturer's instructions; COXI is mtDNA-encoded protein and SDH-A is nDNA-encoded protein. The COXI protein synthesis relative to SDH-A synthesis is the quantitative measurement for mitochondrial biogenesis and content.

**TUNEL assay.** Samples underwent the TUNEL assay using Click-it™ TUNEL Alexa Fluor™ 488 Imaging Assay (#C10617, Thermo Fisher) to evaluate mtDNA breaks. MTR was first stained for visualizing mitochondria, followed by fixation. The processes of TdT incorporation of EdUTP into dsDNA strand breaks and incubation with fluorescent dye detecting the EdUTP were conducted according to the manufacturer's instructions. Colocalization between MTR and TUNEL fluorescence was visualized to detect mtDNA damage, incomplete digestion of mtDNA by mitophagy impairment, reflecting inversely proportional to the mitophagy levels[30]. Measuring TUNEL fluorescence intensity in mitochondria was performed using Fiji software.

**Fluorescence imaging of synaptic vesicle labeling and recycling in hippocampal neuron.** For measuring recycling synaptic vesicle, FM4-64 dye (#T3166, Thermo Fisher) and synaptotagmin-1 antibody were used following the previous study[64]. Hippocampal neurons were loaded with FM4-64 dye and stimulated with high K+-solution. Images were taken before and after destaining with high K+-solution with Eclipse Ts2™ fluorescence microscopy (Nikon, Tokyo, Japan) and underwent analysis of destaining kinetics using Image J. For vesicle labeling, hippocampal neurons were stimulated with depolarization buffer in the presence of synaptotagmin-1 antibodies in neurobasal media. After several washes, neurons were fixed and underwent immunocytochemistry.

**Annexin V-FITC/PI staining.** Annexin V-FITC and PI staining was performed with an Annexin V-FITC apoptosis detection kit (#BD 556547, BD Bioscience, Franklin Lakes, NJ, USA). The analysis was performed following the manufacturer's instructions. After treatment, SH-SY5Y cells were suspended in binding buffer. Then Annexin V-FITC and PI were added to the samples and incubated for 15 min at room temperature. Apoptosis of the samples was detected with flow cytometry (Quanta SC; Beckman Coulter), and data were collected with CytExpert 2.3 software provided from Beckman Coulter. Annexin V-FITC only positive cells

undergo early apoptosis, whereas PI-positive only cells undergo necrosis. Both Annexin V-FITC and PI-positive cells undergo late apoptosis.

**Mitochondrial stress test assay**. OCR under mitochondrial stress test assay was performed using the XF Cell Mito Stress Test Kit (#103015-100) and Seahorse XF24 Extracellular Flux Analyzer (Agilent Technologies, Santa Clara, CA, USA), following the manufacturer's instructions. Hippocampal neurons and SH-SY5Y cells were cultured in XF24 cell culture microplate. Oligomycin, FCCP, and antimycin A/rotenone mixture were treated to cell culture in order for determining the mitochondrial respiration including basal/maximal respiration, proton leak, and ATP production.

**Chromatin immunoprecipitation**. ChIP assay was performed by using an EZ-ChIP Chromatin Immunoprecipitation Kit (#17-371, EMD Millipore, Burlington, MA, USA) according to the manufacturer's instructions. Samples including protein-chromatin complexes were incubated with ChIP grade antibody for GR, the normal IgG, and the RNA polymerase (RNAPol) overnight at 4 °C. Normal IgG and RNAPol were used as negative and positive controls, respectively. Sample DNA was extracted by supplied column and amplified by PCR using a designed primer. The sequences of *PPARGC1A* primer are as follows: forward primer, 5′-GAAAAATAG GAGCCGGGAAT-3′ and reverse primer, 5′-CCGAAGAGTTGCTGCAGTTT-3′. One percent of the sample chromatin extract was used as an input.

**Experimental design of animal study**. Male ICR mice exposed to corticosterone were used to explore the effect of stress-induced corticosterone levels on mitophagy, synaptic homeostasis, and subsequent behavior changes. The hippocampus of mice was mainly used for evaluating glucocorticoid effect on mitophagy, as the hippocampus has the most abundant GRs among the brain. Male ICR mice aged 7 weeks were used, in compliance and approval with the Institutional Animal Care and Use Committee of Seoul National University (SNU-190917-6). Mouse aged 6–8 weeks is widely used to observe the pathological changes leading to neurodegeneration with pharmacological handling[65]. These relatively young mice are suitable for detecting early features of neurodegeneration such as mitochondrial dysfunction, because age-related hippocampal damage exacerbates glucocorticoid-induced neurodegenerative effect, which can interfere the results[66]. Animals were housed six per cage under standard environmental conditions (22 °C relative humidity 70%; 12 h light: dark cycle; ad libitum access to food and drinking solution). Total ninety of 7-week-old male ICR mice were used for the in vivo study. Applying size of samples (minimum of $n = 3$) can be acceptable if very low $p$ values are observed rather than the large size of $n$ including interfering results. Therefore, we set the minimum of $n = 3$ (western blotting, PCR array) and $n = 5$ (behavior test) to gain statistical powers according to the previous published article. The experiments were designed in compliance with the ARRIVE guidelines. Allocations of animals were randomly done to minimize the effects of subjective bias.

As mentioned, 350 ng/ml in serum is regarded as the stress-induced levels of corticosterone on mice. Following the reference demonstrating serum from mice with 10 mg/kg injection reach 350 ng/ml and saturate the GR for most of the day, we set 10 mg/kg corticosterone as high levels of corticosterone[67]. In contrast, 1 mg/kg corticosterone is observed to induce physiological levels of corticosterone in mice[68]. Thus, we set 1 mg/kg and 10 mg/kg corticosterone as low levels and high levels of corticosterone, respectively, to differentiate the effects of physiological and stress-induced corticosterone levels on mitophagy. Corticosterone (1 mg/kg and 10 mg/kg) was dissolved in the solution containing 50% propylene glycol in PBS and injected intraperitoneally daily until killing[5]. Vehicle-treated mice were similarly injected with the solution containing propylene and PBS. PMA (200 µg/kg) was dissolved in the solution containing 1% dimethyl sulfoxide (DMSO) and 99% corn oil (Sigma Chemical Company). The dosage and treatment period of PMA are modified from previous reports[69]. Single intraperitoneal injection of PMA was performed at the starting day. RU 486 (5 mg/kg) was dissolved in the solution containing ethanol and injected daily until killing[70]. Vehicle-treated mice were similarly injected with the solution containing propylene glycol, ethanol, and DMSO. Mice were monitored twice a day during all experiments.

**Y-Maze spontaneous alternation test**. Y-maze spontaneous alternation test is dependent on the innate nature of rodents to differently explore new environments. This behavior test is widely used for quantifying the spatial memory of the rodents. Rodents usually prefer to challenge a new arm of the Y-maze rather than returning back to the one which was previously explored. Before the test, the animals were placed in the home cage at the testing room for 3 h to minimize the stress. The mice were then allowed for 10 min to explore the Y-shaped maze purchased from Sam-Jung Company (Seoul, Korea), whereas the number of arm entries and triads were recorded to calculate the percentage of alternation. Only an entry when all four limbs were within the arm was counted. The alternation amount represents the number of alternations which was divided by total triads (total entries-2). When the animals show higher alternation percentage, the animals tend to maintain the spatial memory function well.

**Forced swim test**. Forced swim test is a rodent behavior test to evaluate the depressive-like behavior. Mice were subjected to a forced swim test for 6 min in a beaker (15 cm × 20 cm) filled with tap water at room temperature, and the trials

were analyzed by Smart 3.0 video tracking system. Generally, only the last 4 min of the test are analyzed owing to the fact that most mice are very active at the beginning of the test. Immobility was defined as floating or remaining motionless without leaning against the wall of the cylinder. When the animals show more immobility, the animals tend to suffer the depression.

**Immunohistochemistry**. Mice underwent deep anesthesia with zoletil (50 mg/kg) and were perfused transcardially with calcium-free Tyrode's solution followed by 4% paraformaldehyde. The removed brain was post-fixed for 2 h in 4% paraformaldehyde and subsequently dehydrated in 30% sucrose in PBS for 24 h at 4 °C. Serial transverse sections (40 µm) were performed using a cryostat (Leica Biosystems, Nussloch, Germany). The brain tissues containing hippocampus were fixed with 4% paraformaldehyde and then pre-blocked with 5% NGS containing 0.1% Triton X-100 in PBS at room temperature for 1 h. Samples were incubated with primary antibodies overnight at 4 °C and then followed by secondary antibodies for 2 h at room temperature. Completed samples were visualized by SRRF imaging system. The fluorescent intensity analysis and acquirement of Pearson's correlation coefficient values were undertaken using Fiji software.

**Statistics and reproducibility**. Imaging experiments and animal tests were conducted and assessed in a blinded fashion. Sample sizes were kept similar between experimental groups and replicates of experiments. The sample size $n$ represents the number of biological independent replicates and statistical analyses were performed using these independent values. The unpaired student's $t$ test was conducted to compare the means of the treatment groups with that of the control group. One-way ANOVA (with Dunnett's multiple comparison test) or two-way ANOVA (with Tukey's multiple comparison test) was used for analyzing the differences among multiple groups. For measuring colocalization levels in images, the values of Pearson's correlation coefficient were obtained from images of each treatment group, and unpaired student's $t$ test and ANOVA were done to confirm whether changes were statistically significant. Results are expressed as mean value ± standard error of mean and analyzed with the GraphPad Prism 6 software (Graphpad, CA, USA). A result with a $p$ value of <0.05 was considered statistically significant and the exact $p$ values for each figure were provided as the table (Supplementary Table).

**Reporting summary**. Further information on research design is available in the Nature Research Reporting Summary linked to this article.

## Data availability
The authors declare that all the data supporting the findings of this study are available within this article, its supplementary information files, or are available from the corresponding author, who has all relevant data, upon reasonable request. Source data are provided with this paper.

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

## Acknowledgements

This research was supported by National R&D Program through the National Research Foundation of Korea (NRF) funded by the Ministry of Science, ICT & Future Planning (NRF-2020R1A2B5B02002442). We appreciate Eun Mi Hur from Seoul National University for her careful advice on our manuscript.

## Author contributions

G.E.C. contributed to the design of the work, performance of experiments, interpretation of data, and writing the paper. H.J.L. contributed to design of the work and interpretation of data. C.W.C., J.H.C., Y.H.J., J.S.K., S.Y.K., and J.R.L. contributed to performance of experiments. H.J.H. contributed to design of the work, interpretation of data, and writing the paper.

## Competing interests

The authors declare no competing interests.
