## [Peer Review File · Nature Communications]

REVIEWER COMMENTS

Reviewer #1 (Remarks to the Author):

In this manuscript, the authors describe a novel regulation system of NIX-mediated mitophagy in neurons. During stress response, glucocorticoid binding to its receptor inhibits PGC1 α activity and eventually downregulates NIX activity, leading to reduced mitophagy, mitochondrial biogenesis and function and, ultimately, synaptic defects. This paradigm is explored both in vitro, using hippocampal neurons and a neuroblastoma cell line, and in vivo using a stress-induced mouse model. Finally, genetic (GR knockdown, PGC1 α and NIX overexpression) and pharmacological (PMA, known NIX enhancer) reversed the effects of glucocorticoid. Although the data is of potential interests there are many methodological concerns that should be addressed to support the author's claims. For example, the mitophagy analysis should be improved, why do they use so many different readouts that do not really determine mitophagy? Similar with mitochondrial biogenesis. The images displayed through the manuscript are of very bad quality and nothing can be really seen. They conclude that glucocorticoid affect only basal mitophagy. What does this mean? How is this demonstrated?

Major points that need to be addressed:

In Figure 1a, it is really striking the lack of Tau staining in corticosterone-treated neurons. Is the image representative? Does corticosterone also reduce its expression or distribution? Is the graph shown depicting co-localization of MTR-Tau or MTR-MAP2?

In Figure 1b, how do the authors quantify mitochondrial distribution (nuclear-synapsis)? The axis states puncta but the legend refers to Pearson's correlation, please clarify.

In Figure 1c, the authors analyze "mitochondria-nucleus" co-localization, however if performed correctly (z-step by z-step), there should be no such thing such as mitochondria-nucleus co-localization. The authors may want to reconsider this measurement and use either mitochondrial area or distance from the nucleus.

In Figure 2i, the authors state that there is a decrease in mitophagy by measuring TOMM20 levels. Whilst mitochondrial mass is a good indicative of such phenomenon, it should also be performed in the presence of lysosomal inhibitors (such as Bafilomycin) to assess mitophagy flux. Could the authors supply the aforementioned data? Please refer to: Kliensky DJ, Abdelmohsen K, Abe A, et al. Guidelines for the use and interpretation of assays for monitoring autophagy (3rd edition).

The annexin V PI flow cytometry dot plots are not compensated making the data difficult to understand (Figure 1i)

Positive controls for mitophagy would strengthen the results, for example in figure 2a.

Again, the images in figure 2 are difficult to interpret. Where are the nucleus in figures 2a and b? Why the cell in figure 2b in the control treatment has so funny shape? Is this representative?

In Supp Figure 2b, the authors assess mitophagy solely by mtDNA:nDNA ratio, the authors may provide additional assays (mtKeima, WB) to support the claim that antimycin A1 does not interact nor induce mitophagy upon cortisol treatment. The same thing stands for Figure 3k, Figure 5c-5d and Figure 6g.

Could the author's explain how citrate synthase activity is indicative of mitochondrial biogenesis? Why do they also use MitoBiogenesis kit?

mtDNA is not a way of measuring mitophagy, this should be changed, line 153.

In Figure 7a, while LC3:TOMM20 co-localization is indicative of mitophagosome formation, it does not necessarily imply that there is fusion into mitophagolysosome and functional mitophagy. Could the authors also assess effective mitochondrial delivery to lysosomes with a lysosomal staining such as LAMP1/2?

Minor points

- Overall, the English throughout the manuscript is very deficient. There are grammatical mistakes that may lead to misunderstanding of the data presented (eg. line 262, 374). Please revise and, if possible, run it through with a native/proficient speaker.
- There are many references that are missing for example:
 - Line 1 in relation of glucocorticoid and AD
 - Line 67-69, there is two missing references regarding epinephrine, deficient mitophagy in neurons and its contribution to AD, please include.
 - Line 75-77
 - In line 81, the authors might want to refer alternatively to the following publication that better illustrates their claim:
 - o McWilliams TG, Prescott AR, Montava-Garriga L, et al. Basal Mitophagy Occurs Independently of PINK1 in Mouse Tissues of High Metabolic Demand. *Cell Metab.* 2018;27(2):439-449.e5. doi:10.1016/j.cmet.2017.12.008
 - In line 120, there is a missing reference regarding mitochondrial mis-localization and ATP production and calcium buffering, please include it in the manuscript.
 - In line 179, the authors claim that mt-Keima induces PARK2 overexpression, could the authors provide a reference regarding this issue?
 - In line 244, there is a missing reference regarding PGC1 as a putative therapeutic target in neurodegenerative diseases.
- Some references do not refer/describe what the authors mention, for example reference 3, in independent of mitophagy.

Reviewer #2 (Remarks to the Author):

The work by Choi and colleagues is interesting and timely. Mitochondrial dysfunction in neural tissue is a key factor that induces many neurodegenerative diseases. The authors use in both in vitro and in vivo corticosteroid treatment to investigate the effect of corticosteroids on mitophagy dysfunction. The authors demonstrate that reduced NIX is closely associated with stress-induced dysfunctional mitophagy, an important in vitro finding in neurons, but not enough to suggest a treatment approach for neurodegenerative diseases. While the overall study is important there are several key issues that must be addressed. Overall, there is a conceptual gap between in vitro corticosteroid administration to neurons and the stress-induced response in vivo. The in vivo stress-induced response affects multiple systems and cells, and not only neurons; thus, it is not clear to what extent the in vitro findings will be applicable to living organisms. Also, for their in vivo studies, the authors used 7-week-old mice and extrapolate to neurodegenerative diseases, which are mainly associated with aging. These young mice therefore may not be an optimal model. The authors need to tone down their conclusion regarding implications to disease, mainly in the Abstract.

Specific comments:

1. For all the in vitro studies, it is stated than n=5. What is this n? Do the authors mean that 5 coverslips were used? If so, it is important to know whether the 5 coverslips come from a single

experiment. How many experimental replications were performed?

2. An experiment showing dose dependence should be added, to substantiate the claim that dose and duration of exposure to corticosteroids are important in coping with stress.

3. The in vivo studies should be performed testing different dosages. In addition, based on the in vivo studies, it is not possible to conclude that the effect on the neurons is directly mediated by the corticosteroids.

Reviewer #3 (Remarks to the Author):

The main findings of this study are that glucocorticoid exposure reduces mitophagy, and impairs mitochondrial and synaptic function in both mice and human cell culture. These effects were shown to be mediated via an activation of the glucocorticoid receptor and a reduction of the NIX pathway. Activation of this pathway prevented the effect of glucocorticoid exposure on these functions as well as on spontaneous alternation behavior in a Y-maze. The effects appear to be robust and convincing.

The authors interpret glucocorticoid effects as being entirely maladaptive. According to the first sentence of the introduction, stress-induced glucocorticoid release might even result in Alzheimer's Disease. However, glucocorticoid release during acute stressful conditions serves many highly adaptive physiological processes. Chronic elevation of glucocorticoid release may however have some maladaptive outcome. Further, the context of glucocorticoid release is important. Glucocorticoid release during a stressful condition induces entirely different effects than pharmacological administration of glucocorticoids without any stress. This should be made much clearer. Throughout the manuscript, the authors do not mention the dose and duration of glucocorticoid exposure. (I was able to find the concentration in the methods, but was unable to find for how long the cells were exposed to the glucocorticoids. Does it resemble an acute or chronic elevation? It seems that the dose used here, i.e., 1 μ M, is very high, in vitro electrophysiological experiments typically use much lower levels of about 100 nM. In fact, in the discussion the authors mention that another reported that such a lower dose of corticosterone actually improves mitochondrial function. Therefore, it would be good if the authors could already state in the introduction and further results section that they are examining the effect of very high (and chronic?) exposure to glucocorticoids. Further, glucocorticoid is not a stress hormone but class of hormones. Therefore, the authors should not say that they exposed the cell cultures to glucocorticoid but to corticosterone or cortisol. It is also not correct to state that the glucocorticoid receptor is mainly activated by pathophysiological levels of glucocorticoids. They become activated during stress but also during the diurnal peak of glucocorticoid release (which, as mentioned above, both serve highly adaptive functions).

The authors made excessive use of words to indicate that all effects found were maladaptive and dysfunctional. As an example, just in the Abstract they mention: ...damaged mitochondria..., ...synaptic defects..., ...mitophagy dysfunction..., ...mitochondrial mislocalization..., ...impaired membrane potential..., ...glucocorticoid-induced damage..., glucocorticoid hampered mitophagy... and ...stress-induced dysfunctional mitophagy. These are clear functional interpretations of the findings. It would be more correct to simply present the effects and then in the discussion interpret these findings in terms of adaptive or maladaptive changes. However, this can actually be challenging as even apparently dysfunctional changes could sometimes be adaptive. For example, glucocorticoids can reduce hippocampal function during stressful conditions to switch toward more goal-oriented and habit-like behavioral responding.

The authors place the effects of glucocorticoid exposure on reduced mitophagy in the context of neurodevelopmental disorders. Would it not be more logical to explain this in the context of development of neuropsychiatric disorders, e.g. major depression, that are associated with chronic elevations of cortisol and reduced hippocampal function? Further, the effects might be relevant to clinical conditions with high-dose glucocorticoid treatment to reduce e.g. inflammatory reactions but

that could have neuropsychiatric and behavioral side effects.

The behavioral experiment is not very well explained. The authors refer to this as a stress-induced mouse model, but in fact the animals were treated with glucocorticoids and not exposed to a stressor. It is not also clear whether this was a single drug administration or whether multiple doses were used. Further, in the main text they only mention that it impaired memory but did not mention the specific memory task.

Reviewer #1 (Remarks to the Author):

In this manuscript, the authors describe a novel regulation system of NIX-mediated mitophagy in neurons. During stress response, glucocorticoid binding to its receptor inhibits PGC1 α activity and eventually downregulates NIX activity, leading to reduced mitophagy, mitochondrial biogenesis and function and, ultimately, synaptic defects. This paradigm is explored both in vitro, using hippocampal neurons and a neuroblastoma cell line, and in vivo using a stress-induced mouse model. Finally, genetic (GR knockdown, PGC1 α and NIX overexpression) and pharmacological (PMA, known NIX enhancer) reversed the effects of glucocorticoid. Although the data is of potential interests there are many methodological concerns that should be addressed to support the author's claims. For example, the mitophagy analysis should be improved, why do they use so many different readouts that do not really determine mitophagy? Similar with mitochondrial biogenesis. The images displayed through the manuscript are of very bad quality and nothing can be really seen. They conclude that glucocorticoid affect only basal mitophagy. What does this mean? How is this demonstrated?

- As suggested by reviewer, the data from the mtDNA damage assay and TUNEL assay have been replaced with those from alternate methods (mt-Keima transfection or western blotting) to improve the analysis of mitophagy levels. Furthermore, we used only the MitoBiogenesis™ in Cell ELISA kit purchased from abcam (#ab140359) to measure mitochondrial biogenesis to avoid overlapping methodologies. We have also replaced all poor quality images which the reviewer indicated with higher quality images. Finally, we have addressed the interpretation of our observations that glucocorticoid primarily affected basal mitophagy in the Discussion section. We also did our best to demonstrate the PINK1-parkin pathway has an insignificant role in inhibitory effects of glucocorticoid on NIX-mediated mitophagy with various experiments.

Major points that need to be addressed:

1. In Figure 1a, it is really striking the lack of Tau staining in corticosterone-treated neurons. Is the image representative? Does corticosterone also reduce its expression or distribution? Is the graph shown depicting co-localization of MTR-Tau or MTR-MAP2?

- The reviewer mentioned that immunofluorescence intensity of Tau seems lower than that of the control group, and that the quality of this image is poor. Therefore, the reviewer questioned whether corticosterone had a downregulating effect on Tau expression or distribution. We have quantified the intensity of Tau immunofluorescence in all images using Image J software and determined no significant changes in Tau expression between control and corticosterone treatment groups (Figure R1), as mentioned in the Results section. If required, we are willing to include this data analysis as a Supplementary figure. Furthermore, we have replaced the original low-quality data with more representative images to avoid further issues.

The graph mentioned by the reviewer showed both co-localization between MAP2 and MTR, and between Tau and MTR, to indicate that the number of mitochondria reduced by corticosterone in

extremities of both axons and dendrites, respectively. To avoid confusion, we have added a detailed explanation in Results section along with two-channel merged images (MTR with MAP2 and MTR with Tau) in Fig 1a.

Figure R1. Hippocampal neurons were treated with corticosterone (1 μ M) for 24 h. Fixed coverslips were then immunostained with MAP2 (green), Tau (blue), and MTR (red). Representative images are shown in Fig 1a. Tau intensity was quantified using Image J software. Data are presented as means \pm S.E.M. $p = 0.9976$. $n=5$ from independent experiments.

2. In Figure 1b, how do the authors quantify mitochondrial distribution (nuclear-synapsis)? The axis states puncta but the legend refers to Pearson's correlation, please clarify.

→ Measuring Pearson's correlation coefficient between TOMM20 and DAPI localization may not be appropriate according to the reviewer's comment in question number 3. Thus, we have measured the immunofluorescence intensity of TOMM20 starting from the nucleus until a distance of 20 μ m in the control and corticosterone groups using Image J software. According to the previous research, the perinuclear region surrounds the nucleus up to a distance of approximately 20 μ m [1]. We have also displayed the relative Pearson's correlation coefficient between TOMM20 and synaptophysin compared to the control, which is reflected in the graph of Fig 1b of the original manuscript. Subsequently, we evaluated the changes in mitochondrial distribution in the perinuclear region and synapse upon corticosterone treatment. As requested by the reviewer, we have clarified the quantification method of mitochondrial distribution to avoid confusion in the Results section.

* References

[1] Al-Mehdi, Abu-Bakr, et al. *Science signaling* 5.231 (2012): ra47.

3. In Figure 1c, the authors analyze "mitochondria-nucleus" co-localization, however if performed correctly (z-step by z-step), there should be no such thing such as mitochondria-nucleus co-localization. The authors may want to reconsider this measurement and use either mitochondrial area or distance from the nucleus.

→ According to the reviewer's suggestion, we have revised the method which accurately reflect the

mitochondrial distribution. Similar to Fig 1b, we have set the perinuclear and distal regions about 20 μm from the nucleus and cell extremities, respectively. Next, we evaluated the mitochondrial intensity in these areas and confirmed the changes in the cells under cortisol treatment compared to control cells. The detailed method and the analysis are presented in the Results section and Fig 1c, respectively.

4. In Figure 2i, the authors state that there is a decrease in mitophagy by measuring TOMM20 levels. Whilst mitochondrial mass is a good indicative of such phenomenon, it should also be performed in the presence of lysosomal inhibitors (such as Bafilomycin) to assess mitophagy flux . Could the authors supply the aforementioned data? Please refer to:

Klionsky DJ, Abdelmohsen K, Abe A, et al. Guidelines for the use and interpretation of assays for monitoring autophagy (3rd edition).

- ➔ As the reviewer requested, we have evaluated mitophagic flux using Bafilomycin A1 treatment, following the protocol described in the suggested reference. Analysis and the description of this data are presented in the Results section and Figure Legends (Fig 2i).

5. The annexin V PI flow cytometry dor plots are not compensated making the data difficult to understand (Figure 1i)

- ➔ As the reviewer requested, we have corrected for signal overlap between the different channels of the flow cytometry and replaced the original figure.

6. Positive controls for mitophagy would strengthen the results, for example in figure 2a.

- ➔ As requested by the reviewer, the combination of CCCP and animycin A1 treatment was used to positively induce mitophagy, mentioned in the results section. We have added these results to Figs 2a – 2b. The references for the combination treatment of CCCP and antimycin A1 used as positive controls for mitophagy are included below.

* References

[1] Vives-Bauza, Cristofol, et al. *Proceedings of the National Academy of Sciences* 107.1 (2010): 378-383.

[2] Cummins, Nadia, et al. *The EMBO journal* 38.3 (2019): e99360.

7. Again, the images in figure 2 are difficult to interpret. Where are the nucleus in figures 2a and b? Why the cell in figure 2b in the control treatment has so funny shape? Is this representative?

- ➔ We have merged the existing images of Figs 2a – 2b with the DAPI to reveal the position of nucleus. As the reviewer mentioned that these images were difficult to interpret due to poor quality, we have revised the original data to include more representative images in Figs 2a – 2d.

8. In Supp Figure 2b, the authors assess mitophagy solely by mtDNA:nDNA ratio, the authors may provide additional assays (mtKeima, WB) to support the claim that antimycin A1 does not interact nor induce mitophagy upon cortisol treatment. The same thing stands for Figure 3k, Figure 5c-5d and Figure 6g.

- ➔ As the reviewer requested, we have detected mt-Keima signals using the SRRF system or determined TOMM20 levels using western blotting. The data from newly performed experiments are presented in Supplementary Fig 3b, Figs 3i – 3m, Figs 5a – 5d, and Fig 6g.

9. Could the author's explain how citrate synthase activity is indicative of mitochondrial biogenesis? Why do they also use MitoBiogenesis kit?

- ➔ We used two methods to determine mitochondrial biogenesis to ensure the inhibitory effect of glucocorticoid treatment, including the citrate synthase activity assay kit (#ab119692) and the MitoBiogenesis™ In-Cell ELISA kit (#ab140359). The results from both methods were shown in the original manuscript; however, according to the reviewer comment, we have only retained data from MitoBiogenesis™ In-Cell ELISA kit to avoid redundancy and any controversy during the citrate synthase as a biomarker of mitochondrial biogenesis. Citrate synthase, localized in the mitochondrial matrix, can be used as a marker for mitochondrial biogenesis because it is a rate-limiting enzyme in the first step of the Krebs cycle and a quantitative indicator of oxidative capacity. Thus, citrate synthase activity is a biomarker for mitochondrial density as well as for biogenesis [1 – 3]. However, its levels can differ depending on age and physical activity [4]. Therefore, we have used the commercial test to estimate mitochondrial biogenesis to avoid this issue. The data from the MitoBiogenesis™ In-Cell ELISA kit are shown in Supplementary Fig 2a, and Supplementry Figs 4a – 4b.

* References

- [1] Vincent, G., et al., *Frontiers in physiology* 6 (2015): 51.
- [2] Pellegrin, Maxime, et al. *Scientific reports* 10.1 (2020): 1-14.
- [3] Larsen, S., et al., *The Journal of physiology* 590.14 (2012): 3349-3360.
- [4] Khan, Samiullah, Juliet Roberts, and Shu-Biao Wu. *BMC molecular and cell biology* 20.1 (2019): 3.

10. mtDNA is not a way of measuring mitophagy, this should be changed, line 153.

- ➔ We have fixed the language in this sentence as below, according to the reviewer's suggestion.

“From this observation, we conclude that it is suitable to measure mtDNA levels for detecting mitochondrial contents, which was mainly increased by failure to eliminate dysfunctional mitochondria in our experimental

conditions.”

11. In Figure 7a, while LC3:TOMM20 co-localization is indicative of mitophagosome formation, it does not necessarily imply that there is fusion into mitophagolysosome and functional mitophagy. Could the authors also assess effective mitochondrial delivery to lysosomes with a lysosomal staining such as LAMP1/2?

- As the reviewer suggested, we performed IHC to support mitochondrial delivery to lysosomes. The brain tissue was immunostained using LAMP1 (lysosomal) and TOMM20 (mitochondrial) antibodies. The representative images and graphs depicting co-localization are shown in Supplementary Fig 5b.

Minor points

1. Overall, the English throughout the manuscript is very deficient. There are grammatical mistakes that may lead to misunderstanding of the data presented (eg. line 262, 374). Please revise and, if possible, run it through with a native/proficient speaker

- As the reviewer requested, we have corrected all grammatical errors and the manuscript has been proofread by a native English speaker. The certificate of English proofreading is displayed below.

2. There are many references that are missing for example

- Line 1 in relation of glucocorticoid and AD

- Line 67-69, there is two missing references regarding epinephrine, deficient mitophagy in neurons and its contribution to AD, please include.

- Line 75-77

- In line 81, the authors might want to refer alternatively to the following publication that better illustrates their claim:

o McWilliams TG, Prescott AR, Montava-Garriga L, et al. Basal Mitophagy Occurs Independently of PINK1 in Mouse Tissues of High Metabolic Demand. Cell Metab. 2018;27(2):439-449.e5. doi:10.1016/j.cmet.2017.12.008

- In line 120, there is a missing reference regarding mitochondrial mis-localization and ATP production and calcium buffering, please include it in the manuscript.

- In line 179, the authors claim that mt-Keima induces PARK2 overexpression, could the authors provide a reference regarding this issue?

- In line 244, there is a missing reference regarding PGC1 as a putative therapeutic target in neurodegenerative diseases.

- Some references do not refer/describe what the authors mention, for example reference 3, in independent of mitophagy.

- We have added the appropriate references or explanation per the reviewer's suggestions.

CERTIFICATE OF EDITING

This is to certify that the paper titled **NA** commissioned to us by **Geu Euhn Choi** has been edited for English language, grammar, punctuation, and spelling by Enago, the editing brand of Crimson Interactive Korea & Co. Ltd. under Advance Editing.

- ✓ **ISO 17100:2015**
Translation Service Providers
- ✓ **ISO 27001:2013**
Information Security Management System
- ✓ **ISO 9001:2015**
Quality Management System

Issued by
Crimson Interactive Korea Co., Ltd.
775, Gyeongin-ro, Yeongdeungpo-gu, Seoul
Ace High Tech City 1-dong 803-74 07299

Disclaimer: The intent of the author's message has been preserved during the editing process. The author is free to accept or reject our changes in the document after reviewing our edits. This certificate has been awarded at the time of sharing the final edited version (full file or sections of the file) with the author. Enago does not bear any responsibility for any alterations done by the author to the edited document post **23 Oct 2020**.

Japan www.enago.jp, www.ulatus.jp, www.voxtab.jp
Taiwan www.enago.tw, www.ulatus.tw
China www.enago.cn, www.ulatus.cn
Brazil www.enago.com.br, www.ulatus.com.br
Germany www.enago.de

Russia www.enago.ru
Arabic www.enago.ae
Turkey www.enago.com.tr
S. Korea www.enago.co.kr
Global www.enago.com, www.ulatus.com, www.voxtab.com

About Crimson:
Crimson Interactive LTD is one of the world's leading academic research support services. Since 2005, we've supported over 2 million researchers in 125 countries with their publication goals.

Reviewer #2 (Remarks to the Author):

The work by Choi and colleagues is interesting and timely.

Mitochondrial dysfunction in neural tissue is a key factor that induces many neurodegenerative diseases. The authors use both *in vitro* and *in vivo* corticosteroid treatment to investigate the effect of corticosteroids on mitophagy dysfunction. The authors demonstrate that reduced NIX is closely associated with stress-induced dysfunctional mitophagy, an important *in vitro* finding in neurons, but not enough to suggest a treatment approach for neurodegenerative diseases.

While the overall study is important there are several key issues that must be addressed.

Overall, there is a conceptual gap between *in vitro* corticosteroid administration to neurons and the stress-induced response *in vivo*. The *in vivo* stress-induced response affects multiple systems and cells, and not only neurons; thus, it is not clear to what extent the *in vitro* findings will be applicable to living organisms. Also, for their *in vivo* studies, the authors used 7-week-old mice and extrapolate to neurodegenerative diseases, which are mainly associated with aging. These young mice therefore may not be an optimal model. The authors need to tone down their conclusion regarding implications to disease, mainly in the Abstract.

- ➔ As the reviewer mentioned, this study focused on the detrimental effect of glucocorticoids on mitophagy and subsequent synaptic defects, and not necessarily on finding a treatment for neurodegenerative diseases. Discovering a treatment for neurodegenerative disease can be an exaggerated interpretation of our study, so we minimized our discussion on the potential therapeutic strategies for clinical use throughout the revised manuscript to avoid confusion.

The reviewer has commented that there is a conceptual gap between a stress-induced mouse model and a corticosterone-exposed mouse model. Stress has systemic effects that the discrepancy between two models may occur, even though stress also leads to the secretion of glucocorticoids; corticosterone injection into mice is typically used to determine stress-induced changes *in vivo*. Many studies use corticosterone to trigger behavioral changes such as anxiety/depression-like behavior or cognitive impairment, and neuronal atrophy including synaptic defects, which are also induced by stress [1, 2]. In contrast, stress-induced mouse models vary depending on the duration and types of stressor. Most studies use a chronic unpredictable stress protocol or give restraint stress to rodents, which become one type of stress-induced mouse [3, 4]. However, the corticosterone injection cannot be considered equivalent to the stress-induced mouse model, which can also trigger other stress-induced factors such as corticotropin-releasing hormone (CRH). Thus, we have changed the text to read “the corticosterone-exposed mouse model.” and avoided the exaggeration of the interpretation of the results.

Inevitable differences occur when applying any *in vitro* experimental results to *in vivo* models. Changes in glial homeostasis by glucocorticoid exposure may also affect neuronal mitophagy in *in vivo* models of this study; glial changes such as activation of astrocytes or microglia can affect neuronal mitophagy. Our hippocampal neuron culture system was supported by glia, as neurons are dependent on trophic support

from glial cells in the hippocampus. Furthermore, we confirmed that transcellular mitophagy by astrocytes was not altered upon corticosterone treatment. Collectively, we have established the optimal *in vitro* and *in vivo* models to minimize this difference as much as possible, based on previously published research.

Many studies use relatively young mice (6–8 weeks) to show pathological changes leading to neurodegeneration with pharmacological or genetic manipulation [5, 6]. The purpose of this study was to investigate the early features of neurodegeneration such as impaired mitochondrial function, neurogenesis, and altered responsiveness to hormones prior to accumulation of toxic molecules such as A β and neurofibrillary tangles, which are usually only found in the brains of aged rodents [7]. Furthermore, age-related hippocampal damage exacerbates glucocorticoid-induced neurodegeneration, which could interfere with our results [8]. Our *in vivo* models were sufficient to determine the effect of corticosterone exposure on mitophagy and subsequent synaptic defects. However, we do not claim the interpretation that these changes directly induce neurodegenerative diseases such as AD, which may need more time to be triggered as the reviewer mentioned. We described the rationale for using 7-week-old mice to observe the effect of glucocorticoid treatment on mitophagy in the Methods section.

Finally, we revised the conclusion regarding the implications toward neurological disorders in Abstract.

*References

- [1] Mitra, Rupshi, and Robert M. Sapolsky. *Proceedings of the National Academy of Sciences* 105.14 (2008): 5573-5578.
- [2] Zhao, Yunan, et al. *Brain research* 1261 (2009): 82-90.
- [3] Wolf, Gilly, et al. *Translational psychiatry* 8 (2018): 124.
- [4] McGill, Bryan E., et al. *Proceedings of the National Academy of Sciences* 103.48 (2006): 18267-18272.
- [5] Howell, Kristy R., Ammar Kutiyawalla, and Anilkumar Pillai. *PloS one* 6.5 (2011): e20198.
- [6] Cruz, Jonathan C., et al. *Neuron* 40.3 (2003): 471-483.
- [7] Scopa, Chiara, et al. *Cell Death & differentiation* 27.3 (2020): 934-948.
- [8] Hassan, AHS V., et al. *Experimental neurology* 140.1 (1996): 43-52.

Specific comments:

1. For all the *in vitro* studies, it is stated that $n=5$. What is this n ? Do the authors mean that 5 coverslips were used? If so, it is important to know whether the 5 coverslips come from a single experiment. How many experimental replications were performed?

- ➔ The sample size represents the number of biological replicates (independent experiments), and statistical analyses were performed using these independent values. All independent *in vitro* experiments were performed with two technical replicates. For example, $n=5$ means five independent experiments were performed with two technical replicates each. In this way, we confirmed the reliability of our data. As the reviewer requested, we added the meaning of n in *in vitro* studies in the Figure Legends.

2. An experiment showing dose dependence should be added, to substantiate the claim that dose and duration of exposure to corticosteroids are important in coping with stress.

- ➔ As the reviewer requested, we investigated the dose-dependent effect of corticosterone and cortisol on mitophagy in hippocampal neurons and SH-SY5Y cells, respectively. We observed a dose-dependent effect of corticosterone on synaptic density in hippocampal neurons. We explain the reason for selecting these doses in Results section and the data are shown in Supplementary Figs 1a – 1c.

3. The *in vivo* studies should be performed testing different dosages. In addition, based on the *in vivo* studies, it is not possible to conclude that the effect on the neurons is directly mediated by the corticosteroids.

- ➔ As the reviewer requested, we investigated the dose-dependent effect of corticosterone on mitophagy, expression of synaptic markers, and memory function (Supplementary Figs 5c – 5d). As mentioned in the Results section, 350 ng/ml of corticosterone in serum is approximately equivalent to the stress-induced levels of corticosterone in mice. Previous studies have demonstrated that serum from mice after 10 mg/kg injection of corticosterone reached 350 ng/ml and saturated the GR for the greater part of a day; we therefore set 10 mg/kg corticosterone as high levels of corticosterone [1, 2]. Furthermore, 1 mg/kg corticosterone was observed to approximate physiological levels in mice [3]. Thus, we set 1 mg/kg corticosterone as the level of corticosterone in homeostasis. The rationale for setting the concentration of corticosterone *in vivo* models was stated in the methods section.

As stated above, we used both a stress-induced model and a corticosterone-injected model interchangeably in the original manuscript, which had different interpretations of the data. Thus, we have corrected this error in the revised manuscript, which strengthens the reliability of the *in vivo* model used to explore the effect of corticosterone on mitophagy.

Other effects that can be triggered by the action of glucocorticoids can also indirectly affect neuronal homeostasis, such as changes in glucose metabolism or inflammation. It has been shown that the GR antagonist RU 486 has an inhibitory effect on the signaling mediated by exogenous corticosterone [4]. Our results follow this previous study to show that RU 486 treatment reduced the effect of corticosterone in the hippocampus. Considering that GR is most abundant in the hippocampus among the brain regions and directly bind to the GRE to activate GR, our findings suggest that corticosterone injection likely directly regulates mitophagy in hippocampal neurons [5].

Furthermore, alterations in the interaction between glia and neurons induced by glucocorticoid could affect mitophagy in neurons via neuroregulatory molecules released from activated astrocytes or microglia, as previously mentioned. As a result of these complex interactions, effects of glia can affect neuronal mitophagy. It is possible that mitophagy is reduced by downregulation of NIX in glia upon glucocorticoid exposure; however, we more focused on neuronal mitophagy, as the neurodegenerative effect is maximized in neurons. Unlike non-neuronal cell types, compartmentally restricted mitophagy process in neurons make themselves more sensitive to abnormal mitophagy due to their post-mitotic nature, tremendous demand of ATP, and extraordinary cellular shapes. In addition, there are many studies that introduce glucocorticoids via intraperitoneal injection, subcutaneous injection, or in drinking water to solely investigate the effect in neurons *in vivo* [2]. Behavioral changes such as memory dysfunction and mood disorder are typical phenotypes of neurodegeneration of hippocampus. However, since other cells than neurons were exposed to corticosterone, the *in vivo* results may not be restricted to indicating that corticosterone only affected neuronal mitophagy. Therefore, we state in the Results and Discussion sections that corticosterone injection in the mouse negatively affected mitophagy of brain tissue, and behavior changes were dependent on neurodegenerative phenotypes, triggered by synaptic dysfunction in neurons. We discussed other indirect mechanisms by which mitophagy inhibition might occur upon corticosterone treatment in the Discussion section.

* References

- [1] Carter, Bradley S., David E. Hamilton, and Robert C. Thompson. *Frontiers in neuroscience* 7 (2013): 139.
- [2] Karten, Y. J. G., et al. *Proceedings of the National Academy of Sciences* 96.23 (1999): 13456-13461.
- [3] Maxwell, Christina R., et al. *Neuropsychopharmacology* 31.5 (2006): 897-903.
- [4] Wulsin, Aynara C., James P. Herman, and Steve C. Danzer. *Frontiers in neurology* 7 (2016): 214.
- [5] Meijer, Onno C., J. C. Buurstede, and Marcel JM Schaaf. *Cellular and molecular neurobiology* 39.4 (2019): 539-549.

Reviewer #3 (Remarks to the Author):

The main findings of this study are that glucocorticoid exposure reduces mitophagy, and impairs mitochondrial and synaptic function in both mice and human cell culture. These effects were shown to be mediated via an activation of the glucocorticoid receptor and a reduction of the NIX pathway. Activation of this pathway prevented the effect of glucocorticoid exposure on these functions as well as on spontaneous alternation behavior in a Y-maze. The effects appear to be robust and convincing.

1. The authors interpret glucocorticoid effects as being entirely maladaptive. According to the first sentence of the introduction, stress-induced glucocorticoid release might even result in Alzheimer's Disease. However, glucocorticoid release during acute stressful conditions serves many highly adaptive physiological processes. Chronic elevation of glucocorticoid release may however have some maladaptive outcome. Further, the context of glucocorticoid release is important. Glucocorticoid release during a stressful condition induces entirely different effects than pharmacological administration of glucocorticoids without any stress. This should be made much clearer. Throughout the manuscript, the authors do not mention the dose and duration of glucocorticoid exposure. (I was able to find the concentration in the methods, but was unable to find for how long the cells were exposed to the glucocorticoids. Does it resemble an acute or chronic elevation? It seems that the dose used here, i.e., 1 μ M, is very high, *in vitro* electrophysiological experiments typically use much lower levels of about 100 nM. In fact, in the discussion the authors mention that another reported that such a lower dose of corticosterone actually improves mitochondrial function. Therefore, it would be good if the authors could already state in the introduction and further results section that they are examining the effect of very high (and chronic?) exposure to glucocorticoids.

- ➔ As the reviewer mentioned, there is a difference between stress-induced effects and corticosterone-induced effects in mouse models. Stress induces systemic effects and triggers the release of stress-induced factors such as corticotropin releasing hormone (CRH) or glucocorticoids, to adapt to the stress environment. Most studies describe chronic unpredictable stress or restraint stress protocols as modeling stress-induced mouse [1, 2]. In contrast, injection of high levels of corticosterone is usually used to determine stress-induced changes *in vivo*. Many research use corticosterone to trigger neuronal atrophy and behavior changes, which are also induced by stress [3, 4]. However, despite this discrepancy, we used these two models interchangeably in the original manuscript. We subsequently clarified the use of these *in vivo* models as “corticosterone-induced mouse models” throughout the revised version of manuscript.

We additionally stated the concentrations of corticosterone/cortisol and durations of treatment throughout the study. As mentioned in the Discussion, our experimental condition resembles the chronic elevation of glucocorticoid exposure, which has been shown in our previous publication [5]. In contrast, acute stress

usually lasts for only a few hours according to previous studies [6]. The detailed information about the duration was mentioned in the Figure legends. Furthermore, we tried to set the concentrations of corticosterone and cortisol in such a way to resemble the stress-induced levels of corticosterone in mice, already stated in Methods section of original manuscript but in the Results section of revised manuscript. In this point of view, 1 μ M and 100 nM of glucocorticoid in *in vitro* system are equivalent to high and low concentrations of glucocorticoid in neurodegeneration-related studies, respectively [7]. We newly confirmed that only 1 μ M of corticosterone and cortisol inhibited mitophagy and synaptic density (Supplementary Figs 1a – 1c). Finally, we stated that the purpose of our study was to observe the effect of chronic and high doses of glucocorticoids on mitophagy inhibition and synaptic dysfunction in the Introduction and Results sections.

* References

- [1] Wolf, Gilly, et al. *Translational psychiatry* 8.1 (2018): 1-11.
- [2] McGill, Bryan E., et al. *Proceedings of the National Academy of Sciences* 103.48 (2006): 18267-18272.
- [3] Mitra, Rupshi, and Robert M. Sapolsky. *Proceedings of the National Academy of Sciences* 105.14 (2008): 5573-5578.
- [4] Zhao, Yunan, et al. *Brain research* 1261 (2009): 82-90.
- [5] Choi, Gee Euhn, et al. *Cell death & disease* 9.11 (2018): 1-18.
- [6] Kang, Jae-Eun, et al. *Proceedings of the National Academy of Sciences* 104.25 (2007): 10673-10678.
- [7] Du, Jing, et al. *Proceedings of the National Academy of Sciences* 106.9 (2009): 3543-3548.

2. Further, glucocorticoid is not a stress hormone but class of hormones. Therefore, the authors should not say that they exposed the cell cultures to glucocorticoid but to corticosterone or cortisol. It is also not correct to state that the glucocorticoid receptor is mainly activated by pathophysiological levels of glucocorticoids. They become activated during stress but also during the diurnal peak of glucocorticoid release (which, as mentioned above, both serve highly adaptive functions).

- ➔ As the reviewer requested, we clarified which hormones among glucocorticoids were used in the cell culture and *in vivo* experiments. We also changed the expression used, as reviewer suggested, to “GR is activated mainly by a relatively high level of glucocorticoid”.

3. The authors made excessive use of words to indicate that all effects found were maladaptive and dysfunctional. As an example, just in the Abstract they mention: ...damaged mitochondria..., ...synaptic defects..., ...mitophagy dysfunction..., ...mitochondrial mislocalization..., ...impaired membrane potential..., ...glucocorticoid-induced damage..., glucocorticoid hampered mitophagy... and ...stress-induced dysfunctional mitophagy. These are clear functional interpretations of the findings. It would be more correct to simply present the effects and then in the discussion interpret these findings in terms of adaptive or maladaptive changes. However, this can actually be challenging as even apparently dysfunctional changes could sometimes be adaptive. For example, glucocorticoids can reduce

hippocampal function during stressful conditions to switch toward more goal-oriented and habit-like behavioral responding.

- As the reviewer advised, we have depicted the effects of glucocorticoid treatment on mitochondrial function in the Abstracts and Results sections, trying to exclude the indicated expressions. Efforts were made to faithfully attach meaning to the results themselves as much as possible, so that the interpretation of the results was not exaggerated. We further stated that the interpretation of our experiments was focused on the maladaptive effects of glucocorticoid on mitophagy in the Discussion section reflecting the reviewer's opinion.

4. The authors place the effects of glucocorticoid exposure on reduced mitophagy in the context of neurodevelopmental disorders. Would it not be more logical to explain this in the context of development of neuropsychiatric disorders, e.g. major depression, that are associated with chronic elevations of cortisol and reduced hippocampal function? Further, the effects might be relevant to clinical conditions with high-dose glucocorticoid treatment to reduce e.g. inflammatory reactions but that could have neuropsychiatric and behavioral side effects.

- As the reviewer pointed out, neurodegeneration in the hippocampus contributes to both neurodevelopmental and neuropsychiatric disorders, mainly representing with cognitive impairment and major depression, respectively. Thus, it is necessary to investigate the effects of glucocorticoids on both behavior changes. We additionally performed the forced swim test to determine the effect of corticosterone on depression-like behavior. There was no significant change in depression-like behavior between the experimental groups, shown in Fig 7f. This data was interpreted in the Discussion section.

5. The behavioral experiment is not very well explained. The authors refer to this as a stress-induced mouse model, but in fact the animals were treated with glucocorticoids and not exposed to a stressor. It is not also clear whether this was a single drug administration or whether multiple doses were used. Further, in the main text they only mention that it impaired memory but did not mention the specific memory task.

- As the reviewer requested, we have added a statement describing the purpose of the behavioral experiment in the Results section. The Y-maze was used to explore the spatial memory of rodents, and the forced swim test was used to evaluate depression-like behavior of rodents. Both tests are well-known tools for evaluating the function of the hippocampus, which is responsible for cognitive memory and mood regulation such as in anxiety and depression. In addition, we clarified that the mice exposed to corticosterone specifically, rather than to a stressor. We also included the drug administration schedule and dosage in Methods section and Supplementary Fig 5a. Finally, the memory task referred to throughout the study was the Y-maze test, which is used to assess spatial memory.

REVIEWERS' COMMENTS

Reviewer #1 (Remarks to the Author):

The authors have addressed most of the concerns raised in the original manuscript and provided enough evidence to support their claims. However, some minor changes would help in getting the message across:

- Replace the LC3:TOMM20 co-localization assay in Fig 7a with the new supplementary information in Fig S5b. The ultimate goal of mitophagy is to deliver the cargo (mitochondria) to lysosomes and proceed with its degradation, therefore LC3:LAMP1 assay is more helpful in conveying the idea of mitophagy inhibition by GR activity.

- In line 264, I believe there is a missing word in "to use the innate (?) of rodents".

- The Facs dot plots in figure 1i and Sup4d are not very well compensated, it would be nice repeat the experiment with better compensation so the populations can be easily distinguished without having to draw a diagonal line.

Reviewer #2 (Remarks to the Author):

The authors addressed most of this reviewer's comments in satisfactory way.

Reviewer #3 (Remarks to the Author):

The authors adequately addressed my previous concerns. The only remaining issue is that glucocorticoids should be used in the plural form, thus not as phrased in the abstract 'glucocorticoid inhibits mitophagy' but glucocorticoids inhibit mitophagy.

Reviewer #1 (Remarks to the Author):

The authors have addressed most of the concerns raised in the original manuscript and provided enough evidence to support their claims. However, some minor changes would help in getting the message across:

1. Replace the LC3:TOMM20 co-localization assay in Fig 7a with the new supplementary information in Fig S5b. The ultimate goal of mitophagy is to deliver the cargo (mitochondria) to lysosomes and proceed with its degradation, therefore LC3:LAMP1 assay is more helpful in conveying the idea of mitophagy inhibition by GR activity.

→ As the reviewer suggested, we replaced LC3:TOMM20 co-localization assay with the LAMP1:TOMM20 assay (Fig 8a).

2. In line 264, I believe there is a missing word in “to use the innate (?) of rodents”.

→ We revised the mentioned sentence. “Y-maze test evaluates the spatial memory to use the innate nature of rodents to explore the new objects”

3. The Facs dot plots in figure 1i and Sup4d are not very well compensated, it would be nice repeat the experimnet with better compensation so the populations can be easily distinguished without having to draw a diagonal line.

→ As the reviewer requested, we repeated the experiments and replaced the original data of Fig 1i and Supplementary Fig 4d with more representative images with better compensation and distinguishable populations.

Reviewer #2 (Remarks to the Author):

The authors addressed most of this reviewer's comments in satisfactory way.

Reviewer #3 (Remarks to the Author):

1. The authors adequately addressed my previous concerns. The only remaining issue is that glucocorticoids should be used in the plural form, thus not as phased in the abstract ‘glucocorticoid inhibits mitophagy’ but glucocorticoids inhibit mitophagy.

→ We ensured that glucocorticoids were used in the plural form throughout the manuscript.